# Single-cell transcriptional profiling of splenic fibroblasts reveals subset-specific innate immune signatures in homeostasis and during viral infection

Joern Pezoldt [1,2], Carolin Wiechers[2], Florian Erhard [3], Ulfert Rand [4], Tanja Bulat [5], Michael Beckstette[6], Andrea Brendolan[7], Jochen Huehn [2], Ulrich Kalinke[8], Mathias Mueller [5], Birgit Strobl [5], Bart Deplancke [1], Luka Čičin-Šain [4✉] & Katarzyna M. Sitnik [4✉]

Our understanding of the composition and functions of splenic stromal cells remains incomplete. Here, based on analysis of over 20,000 single cell transcriptomes of splenic fibroblasts, we characterized the phenotypic and functional heterogeneity of these cells in healthy state and during virus infection. We describe eleven transcriptionally distinct fibroblastic cell clusters, reassuring known subsets and revealing yet unascertained heterogeneity amongst fibroblasts occupying diverse splenic niches. We further identify striking differences in innate immune signatures of distinct stromal compartments in vivo. Compared to other fibroblasts and to endothelial cells, Ly6C$^+$ fibroblasts of the red pulp were selectively endowed with enhanced interferon-stimulated gene expression in homeostasis, upon systemic interferon stimulation and during virus infection in vivo. Collectively, we provide an updated map of fibroblastic cell diversity in the spleen that suggests a specialized innate immune function for splenic red pulp fibroblasts.

[1] Laboratory of Systems Biology and Genetics, École Polytechnique Fédérale de Lausanne, 1015 Lausanne, CH, Switzerland. [2] Department of Experimental Immunology, Helmholtz Centre for Infection Research, 38124 Braunschweig, Germany. [3] Institute for Virology and Immunobiology, Julius-Maximilians-Universität Würzburg, 97070 Würzburg, Germany. [4] Department of Viral Immunology, Helmholtz Centre for Infection Research, 38124 Braunschweig, Germany. [5] Institute of Animal Breeding and Genetics, Department of Biomedical Science, University of Veterinary Medicine Vienna, 1210 Vienna, Austria. [6] Department of Computational Biology for Individualized Medicine, Centre for Individualized Infection Medicine, 30625 Hannover, Germany. [7] Unit of Lymphoid Organ Development, Division of Experimental Oncology, IRCCS San Raffaele Scientific Institute, Milan, Italy. [8] Institute for Experimental Infection Research, TWINCORE, Centre for Experimental and Clinical Infection Research, a joint venture between the Helmholtz Centre for Infection Research, Braunschweig, and the Hannover Medical School, 30625 Hannover, Germany. ✉email: luka.cicin-sain@helmholtz-hzi.de; katarzyna.sitnik@vetmeduni.ac.at

The spleen is the largest secondary lymphoid organ (SLO) in the body that plays a non-redundant role in host defence. It emerged alongside with adaptive immunity at the dawn of jawed vertebrate evolution and is the only described bona fide SLO in all vertebrate classes except mammals that additionally have lymph nodes (LNs). The spleen is distinguished into two anatomically and functionally distinct compartments: the red pulp, which is filled with red blood cells and macrophages; and the white pulp, which is comprised of lymphocyte aggregates organized into a T-cell zone and B-cell follicles. The white and red pulp areas are bridged by the marginal zone, which harbours specific subsets of B-cells and macrophages[1,2].

Fibroblastic cells (hereafter referred to as FC) are critical structural and functional components of all SLOs[3]. Splenic FC are best studied in the white pulp, where they have long been known to form three functionally specialized subsets: (i) CCL21-expressing T-zone reticular cells (TRC); (ii) CXCL13- and CR1/2-expressing follicular dendritic cells (FDC), located in the B-cell follicle as well as (iii) CXCL13-, MAdCAM1- and RANKL-expressing marginal reticular cells (MRC)[4]. Our current understanding of splenic FC diversity is based on the analysis of single-cell transcriptomes of 6227 ICAM1[+]/PDGFRβ[+] FC from the spleen of naïve mice originally published by Cheng et al.[5], and further explored by others[6,7]. In addition to corroborating the distinction of white pulp FC into TRC, FDC and MRC, analysis of the above-mentioned dataset distinguished a population of red pulp FC[6]; as well as provided transcriptional evidence for the existence of pericyte-like cells[5] and adventitial cells[7]. Critically, the field still lacks a characterization of splenic FC at single-cell level with a resolution matching the detail with which we currently view the heterogeneity of FC in the LNs, where several FC subsets in addition to FRC, FDC and MRC have been resolved[8,9].

Type I interferons (IFNs) are crucial antiviral cytokines produced by a broad range of, if not by all, cells upon primary contact with viruses. They signal through a heterodimeric transmembrane receptor, IFNα receptor (IFNAR), which is composed of IFNAR1 and IFNAR2 subunits. Binding of type I IFN to IFNAR triggers the Janus kinase/signal transducers and activators of transcription (JAK/STAT) pathway of signal transduction, which ultimately leads to the assembly of a trimeric STAT1/STAT2/IRF9 (ISGF3) transcription factor complex, which enhances the expression of interferon-stimulated genes (ISGs) with distinctive antiviral activities[10,11]. In addition to being produced upon virus infection, type I IFNs are also well recognized for their homeostatic (i.e. tonic) presence and function[12]. Tonic stimulation of cells by type I IFN signalling is thought to raise the cell's antiviral potential by enhancing its responsiveness to acutely produced IFNs[13] and potentially also by upregulating basal expression of antiviral genes[14]. Putative heterogeneity in ISG expression among splenic FC, which may imply potential differences in innate antiviral capacity of distinct FC subsets, remains unexplored.

Here, based on single-cell RNA sequencing of over 20,000 splenic FC, we provide a more detailed view of the phenotypic and functional heterogeneity among fibroblasts from this vital immune organ. Our analysis discerns eleven transcriptionally distinct FC clusters that distribute across four main fibroblastic identities, such as Bst1[+] white pulp fibroblasts, Ly6c1[+] red pulp fibroblasts, Mcam[+] pericytes/vascular smooth muscle cells (VSMC) and Cd34[+] adventitial cells. In addition to confirming the subdivision of white pulp FC into functionally specialized TRC, MRC and FDC subsets, dissection of the intra-population diversity of Bst1[+] FC revealed the existence of functionally specialized Dpt[+]Tnfsf13b[+] FC in the splenic T-cell zone. We also identify heterogeneity among FC localized in the red pulp and in the perivascular niche in the spleen. We further uncover that Ly6c1[+] red pulp fibroblasts express an augmented ISG signature under homeostatic conditions, which is dependent on tonic type I IFN signalling, and upon virus infection in vivo; suggesting a potential specialized role for splenic red pulp stroma in antiviral defence. In sum, our characterization of splenic FC at single-cell level provides insight into the functional diversity of splenic stromal cells.

## Results and discussion

**Single-cell RNA sequencing uncovers vast heterogeneity among splenic fibroblasts.** To study the phenotypic and functional heterogeneity of the splenic fibroblastic cell (FC) compartment, we performed single-cell RNA sequencing (scRNA-seq) of FC from adult mouse spleen. To obtain the full spectrum of FC, we isolated and analysed the total non-endothelial stromal cell fraction identified as CD45[−]ITGB1[+]CD31[−] cells in splenic digests (Fig. 1a and Supplementary Fig. 1a). Note that CD45[−]ITGB1[−] cells represent Transferrin receptor[+] erythroid lineage cells (Fig. 1a). CD45[−]ITGB1[+]CD31[−] cells were FACS-sorted, followed by droplet-based single-cell transcriptome barcoding, reverse transcription and next generation sequencing of the resulting cDNA library (Fig. 1a). After quality control and exclusion of endothelial- (0.5 %) and mesothelial-like cells (2.1 %) as well as putative doublets (1.4 %), we retained for further analysis 21,620 FC with a median of 2730 detected genes per cell (Supplementary Fig. 1b). Following re-embedding, eleven transcriptional clusters were identified and ordered hierarchically using expression intensities of differentially expressed genes (DEGs) with the respective highest fold change per cluster (average log₂(fold change) > 0.25, adjusted p-value < 0.01, fraction of expressing cells 0.1) (Fig. 1b and Supplementary Data 1). Based on overall similarity, the eleven transcriptional clusters were grouped into four fractions: (i) Bst1[+] FC comprising four cell clusters, (ii) Ly6c1[+] FC comprising four cell clusters, (iii) Mcam[+] FC comprising two cell clusters and (iv) a single cluster of Cd34[+] FC (Fig. 1c and Supplementary Fig. 1c). FC clusters belonging to the same fraction were more similar to each other than to any other cluster, indicating they may be subfractions of one main population (Fig. 1d). Consistent with the scRNA-seq data, flow cytometric analysis showed that splenic FC are comprised of four discrete populations distinguished by cell surface expression of protein markers, BST-1 (also known as CD157/BP-3), Ly6C, MCAM and CD34 (Fig. 1e). Neither BST-1[+] nor Ly6C[+] FC were present at birth, but emerged in the first weeks of postnatal life, suggesting that these subsets mature from neonatal precursor population/-s (Fig. 1f). We proceeded to investigate the specific features and intra-population heterogeneity of, respectively, Bst1[+] FC, Mcam[+] FC, Cd34[+] FC and Ly6c1[+] FC.

**Dissection of Bst1[+] FC indicates functional specialization of FC in the splenic T-cell zone.** In keeping with published data on the expression pattern of BST-1 (CD157/BP-3) in the spleen[15], we found that Bst1[+] FC (identified as BST-1[+] cells co-expressing a mesenchymal cell marker, desmin) are selectively localized in the white pulp (Fig. 2a). Consistently, scRNA-seq analysis showed that all Bst1[+] FC clusters expressed white pulp stromal cell marker genes, Mfge8 and Clu[5] as well as regulators of lymphocyte migration and/or differentiation, including Enpp2, a key lysophosphatidic acid (LPA)-producing enzyme linked to T-cell-high speed motility[16]; Il33, a critical immune modulator shaping type 1, type 2 and regulatory T-cell responses[17] and Cxcl9, a chemokine ligand implicated in type 1 T-cell differentiation[18] (Fig. 2b). Of the four Bst1[+] clusters, two corresponded to known white pulp FC subsets, TRC and FDC, respectively represented by the clusters termed Bst1[+]|Ccl21a[+] and Bst1[+]|Cr2[+] (Fig. 2b).

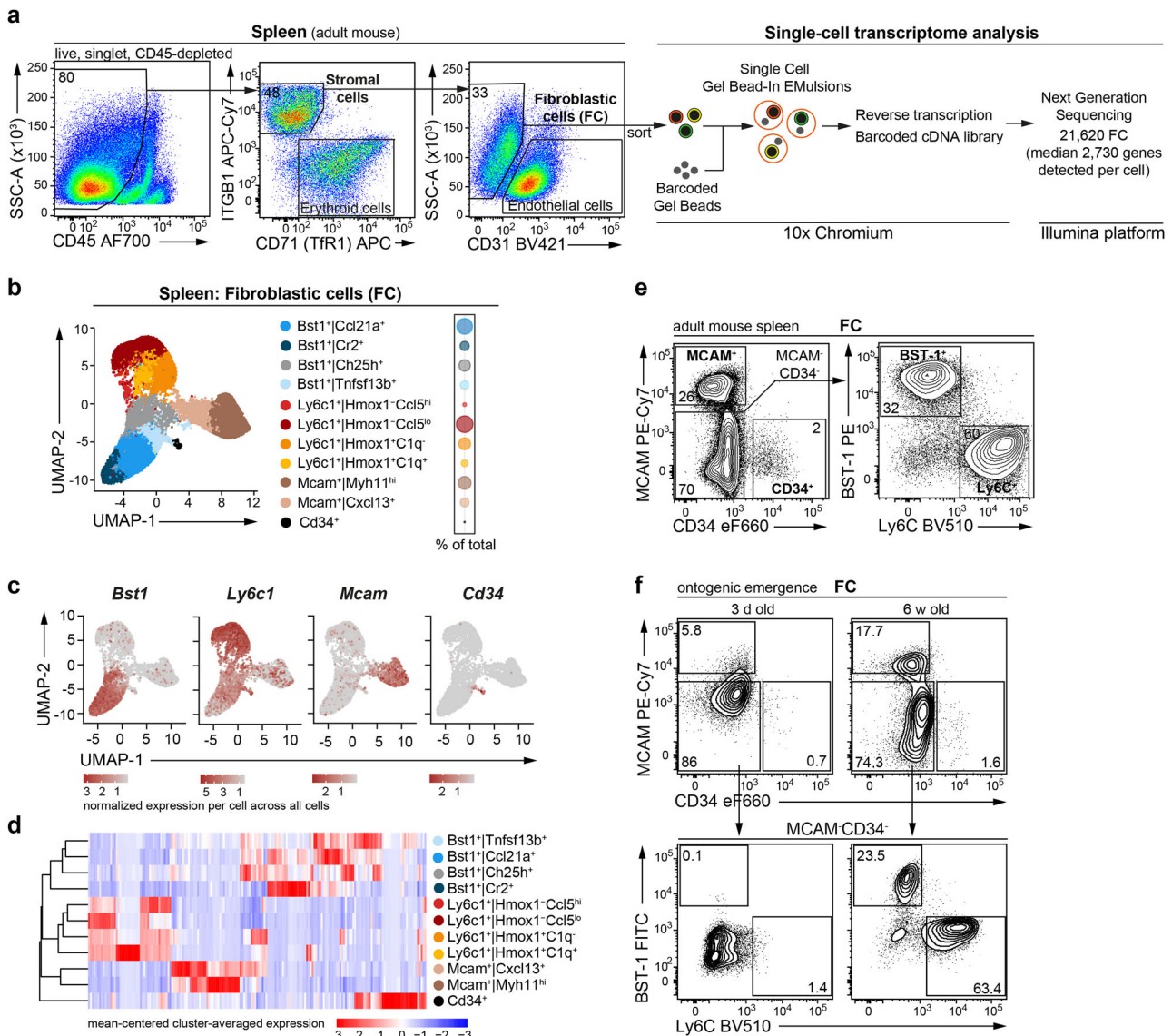

**Fig. 1 Single-cell RNA sequencing uncovers vast heterogeneity among splenic fibroblasts. a** Gating strategy used to isolate fibroblastic cells (FC) from adult mouse spleen (pre-gating shown in Supplementary Fig. 1a) and a flow-chart outlining the steps in scRNA-seq analysis. **b** UMAP embedding of splenic FC. **c** Gene expression on UMAP embedding. **d** Hierarchical clustering of transcriptional clusters based on the mean expression of the top 20 differentially expressed genes (DEGs) per cluster with highest fold change. **a–d** Data pooled from 2 independent experiments using cells sorted from pooled preparations of 3 mice per experiment. **e** Flow cytometric confirmation of *Mcam*⁺ FC, *Cd34*⁺ FC, *Bst1*⁺ FC and *Ly6c1*⁺ FC subsets in the adult spleen. Representative stains from 2 independent experiments with 3 mice per experiment. **f** Emergence of *Bst1*⁺ FC and *Ly6c1*⁺ FC subsets during spleen ontogeny. Representative stains from 2 independent experiments with 3 biological replicates per experiment using pooled cell preparations from 2 spleens/replicate. **a**, **e**, **f** Numbers are percentage of cells in the indicated gates.

Accordingly, the *Bst1*⁺|*Ccl21a*⁺ cluster expressed highest mRNA levels of the T-zone chemokines, *Ccl21* and *Ccl19*, while *Bst1*⁺| *Cr2*⁺ cells expressed the selective FDC-marker, *Cr2* and highest mRNA levels of *Cxcl13* and *Madcam1* (Fig. 2b). Flow cytometric analysis confirmed that *Bst1*⁺ FC selectively contained CCL21⁺ TRC and CR1/2⁺ FDC (Fig. 2c). The third *Bst1*⁺ FC cluster, *Bst1*⁺|*Ch25h*⁺, was comprised of *Ccl19*ˡᵒ*Ccl21a*ˡᵒ*Cxcl13*⁺ cells that selectively expressed *Ch25h* (Fig. 2b). Since *Ch25h* expression marks MRC in the LNs[8] and because the reported localization of *Ch25h*⁺ cells in the spleen (B-cell follicle perimeter)[19] overlaps with the area where splenic MRC are located, we re-embedded *Bst1*⁺|*Ch25h*⁺ FC to determine if they contained MRC-like cells. Indeed, *Bst1*⁺|*Ch25h*⁺ FC encompassed cells expressing MRC-signature genes, *Madcam1* and *Tnfsf11* (encoding RANKL) (Fig. 2d). The remainder of *Bst1*⁺|*Ch25h*⁺ cluster, comprising

*Madcam1*⁻*Tnfsf11*⁻ cells, may be analogous to a previously described *Ch25h*⁺*Ccl19*ˡᵒ*Madcam1*⁻ LN FC cluster located at the LN B-T-cell border[8]. Notably, our analysis also identified a fourth *Bst1*⁺ FC cluster, termed *Bst1*⁺|*Tnfsf13b*⁺ that did not fit TRC, FDC or MRC identity (Fig. 2b), and showed highest expression across all *Bst1*⁺ clusters for *Cxcl12* and *Tnfsf13b* (encoding the B-cell survival factor, BAFF) (Fig. 2b). The *Bst1*⁺|*Tnfsf13b*⁺ cluster was further distinguished from other *Bst1*⁺ FC by the expression of *Dpt* (dermatopontin), a characteristic *Bst1*⁺| *Tnfsf13b*⁺ FC shared with *Cd34*⁺ FC, and by upregulated expression of MHC class II genes (Fig. 2e). Histological examination by in situ RNA hybridization (RNAscope) demonstrated that *Bst1*⁺|*Tnfsf13b*⁺ FC (identified as cells co-expressing mRNAs for *Dpt* and *Tnfsf13b*) reside specifically in the splenic T-cell zone (Fig. 2f). This data also suggested that *Dpt*-expressing

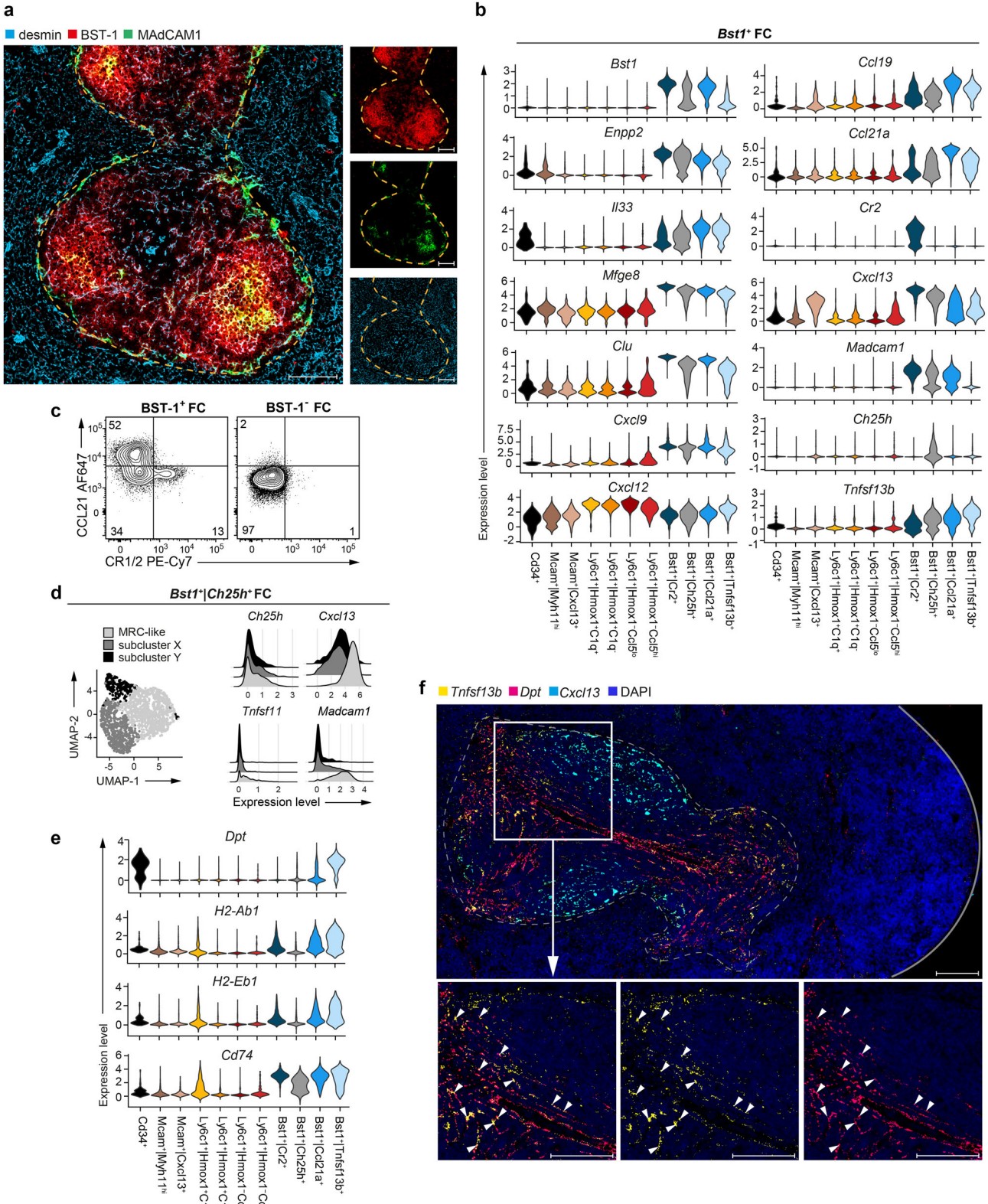

cells constitute a major source of *Tnfsf13b* transcripts in the spleen (Fig. 2f). Notably, B-cell viability and follicular organization in the LNs appear to be maintained by BAFF produced locally by LN FC[20]. Whether *Dpt*+ FC constitute a biologically relevant source of BAFF in the spleen will require additional functional experiments. Collectively, these data resolve additional functional complexity within the FC compartment of

the splenic white pulp that goes beyond subdivision into TRC, FDC and MRC.

**Mcam+ FC and Cd34+ FC represent functionally distinct components of the perivascular niche.** Next, we analysed *Mcam*+ FC and *Cd34*+ FC. *Mcam*+ FC were distinguished from other FC

**Fig. 2 Dissection of *Bst1*+ FC indicates functional specialization of FC in the splenic T-cell zone. a** Confocal microscopy analysis of desmin (light blue), BST-1 (red) and MAdCAM1 (green) demonstrating that *Bst1*+ FC (desmin+BST-1+ cells) are localized in the white pulp. Representative stains from 2 spleens analysed in 2 independent experiments. Dashed yellow line, the white pulp/red pulp border. Scale bars are 100 μm. **b** Violin plots showing expression of marker genes that segregate *Bst1*+ FC (scaled total UMI counts per cell) across individual clusters. **c** Flow cytometric analysis showing that *Bst1*+ FC contain CCL21+ TRC and CR1/2+ FDC. Representative stains from 2 independent experiments with a single biological replicate per experiment using pooled cell preparations from 2 spleens/replicate. Numbers in the gates represent percentage of cells. **d** Scaled gene expression histograms for sub-clusters obtained from re-embedded *Bst1*+|*Ch25h*+ FC. **e** Violin plots showing expression of marker genes that segregate *Bst1*+|*Tnfsf13b*+ FC (scaled total UMI counts per cell) across individual clusters. **b, e** Violin plots were generated using default settings of the function VlnPlot() of the R package Seurat. **f** In situ RNA hybridization (RNAscope) for *Dpt* (pink), *Tnfsf13b* (yellow) and *Cxcl13* (light blue). Sections were counterstained with DAPI (dark blue). Exemplary *Bst1*+|*Tnfsf13b*+ FC (*Dpt*+*Tnfsf13b*+ cells) are indicated by arrowheads. Dashed grey line, the white pulp/red pulp border. Solid grey line, the capsule. Scale bars are 100 μm. Representative stains from 2 spleens analysed in 2 independent experiments.

by selectively upregulated expression of genes associated with pericyte or vascular smooth muscle cell (VSMC)-identity, such as pericyte markers (*Mcam, Rgs5, Notch3, Cspg4* (NG2) and *Esam*); genes associated with VSMC contractility (*Acta2* (ASMA), *Myh11, Myl9, Lmod1, Cald1* and *Tagln* (SM22)) as well as the myocyte lineage marker, *Mef2c*[21,22] (Fig. 3a and Supplementary Data 1). Flow cytometric analysis confirmed the expression of ASMA by *Mcam*+ FC (Fig. 3b). Next, we assessed the localization of *Mcam*+ FC (note *Mcam*+ FC are PDGFRα−, Fig. 3c) by microscopical analysis of MCAM+PDGFRα− cells on splenic sections from *Pdgfra*-H2B-GFP mice, in which the nuclei of PDGFRα+ cells are labelled with GFP. Consistent with the possibility that *Mcam*+ FC are pericytes/VSMC, MCAM+GFP− cells were indeed associated with splenic blood vessels (Fig. 3d, filled arrowheads). Interestingly, *Mcam*+ FC were the main source of neurotrophic factors, such as *Ngf* and *Ntf3* (Fig. 3a), suggesting that *Mcam*+ FC may support nerves which run along intrasplenic arteries and arterioles[23]. Our data also indicated that *Mcam*+ FC are heterogenous, since they encompassed two transcriptional clusters, one expressing relatively higher mRNA levels of *Mcam* and of VSMC markers (*Mcam*|*Myh11*[hi]) and another, with lower expression of *Mcam* and less pronounced contractile profile that expressed higher levels of neurotrophic factors and was marked by *Cxcl13* expression (*Mcam*|*Cxcl13*+) (Fig. 3a). Flow cytometric analysis confirmed the distinction of *Mcam*+ FC into MCAM[hi]CXCL13− and MCAM-[lo]CXCL13+ subsets (Fig. 3e, f), corroborating the scRNA-seq analysis. Next, we assessed the anatomical localization of *Mcam*|*Myh11*[hi] FC and *Mcam*|*Cxcl13*+ FC. To this end, we performed in situ RNA hybridization (RNAscope) for *Notch3*, a pericyte/VSMC marker[21] which robustly discerns both *Mcam*+ clusters from other FC (Fig. 3a), and for *Cxcl13*. In contrast to *Mcam*|*Myh11*[hi] FC (identified as *Notch3*+*Cxcl13*− cells), which were found in both red and white pulp, *Mcam*|*Cxcl13*+ FC surrounded select vessels in the red pulp (Fig. 3g, filled arrowheads). Of note, the *Cxcl13* RNAscope performed in conjunction with the detection of *Dpt* and *Tnfsf13b* shown in Fig. 2f serves only to reveal the positioning of B-cell follicles and should not be used to assess the distribution of *Cxcl13*+ FC due to the markedly lower sensitivity of *Cxcl13* detection relating to the use of a weaker fluorophore and a detection channel with a higher autofluorescence level. As far as we are aware, *Mcam*+*Cxcl13*+ cells have no apparent equivalent among previously described FC populations. Putative specialized function(-s) of the CXCL13+ mural cells in the spleen remain to be addressed by future studies. In contrast to *Mcam*+ FC, *Cd34*+ FC expressed PDGFRα (Fig. 3c) and hence could be identified as CD34+ cells with nuclear GFP signal on splenic sections from *Pdgfra*-H2B-GFP mice (Fig. 3d, empty arrowheads). *Cd34*+ FC formed an outer layer surrounding *Mcam*+ FC around larger splenic vessels, running inside of and close to splenic trabeculae, a localization consistent with perivascular adventitial cells[22,24]. *Cd34*+ FC expressed a gene signature composed of *Igfbp6, Penk, Lum* and *Ptgis*, similar as described for CD34+ fibroblasts in the LNs[8,25]

(Fig. 3h). Further of note, *Cd34*+ FC expressed *Dpt* (Fig. 2e) and *Ly6c1* (Fig. 3h), suggesting they are similar to the recently described population of *Cd34*+*Ly6c1*+*Dpt*+ pan-tissue adventitial cells identified at transcriptional level in multiple tissues[7]. Collectively, the above analysis broadens our current view of the perivascular cell niche in the spleen.

**_Ly6c1_+ FC are located in the splenic red pulp and the marginal zone.** Next, we assessed the distinguishing features of *Ly6c1*+ FC. *Ly6c1*+ FC showed transcriptional similarity to previously described *Csf1*-expressing *Wt1*+ FC[6] and *Cxcl12*-expressing *Tcf21*+ FC[26] of the splenic red pulp. Accordingly, *Ly6c1*+ FC were enriched for mRNAs encoding *Wt1* and *Tcf21*, as well as expressed highest levels of *Csf1* and *Cxcl12* transcripts amongst all FC (Figs. 2b and 4a). *Ly6c1*+ FC were further distinguished by the expression of class B scavenger receptor *Cd36* and of various ECM components and regulators, such as heparan sulphate editing enzyme *Hs6st2* as well as *Npnt* and *Cadm4* (Fig. 4a and Supplementary Fig. 1c). Interestingly, *Ly6c1*+ FC abundantly expressed *Coch* (cochlin), which plays an important role in innate immune defence against bacterial infections[27] (Fig. 4a). Immunohistological staining for a mesenchymal cell marker, desmin and Ly6C demonstrated that Ly6C+desmin+ FC are selectively located in the red pulp and marginal zone (Fig. 4b). Ly6C+desmin+ FC formed a network in-between MECA-32+ sinusoids (Fig. 4b, filled arrowheads) but also appeared to tightly wrap around these vessels (Fig. 4b, empty arrowheads). The existence of *Ly6c1*+ FC positioned in direct contact with MECA-32+ endothelium was also apparent when *Ly6c1*+ FC were visualized as WT1+GFP+ cells on splenic sections from *Pdgfra*-H2B-GFP mice (Fig. 4c, filled arrowheads). Of note, EC are PDGFRα−WT1− (Supplementary Fig. 1d–f). Whereas no apparent diversity within splenic red pulp fibroblasts could previously be discerned[5,6], our analysis distinguished *Ly6c1*+ FC into four clusters with distinctive transcriptional profiles indicating the existence of cell subsets and/or states within the red pulp FC compartment. *Ly6c1*+ FC clusters showed diversity with respect to expression of *Hmox1* (heme oxygenase 1) forming two *Hmox1*− and two *Hmox1*+ clusters (Fig. 4a). The two *Ly6c1*+|*Hmox1*+ clusters were distinguished from each other, respectively, by upregulated complement component 1q expression (*Ly6c1*+|*Hmox1*+*C1q*+) and by the expression of *Gdf15* and of transcripts encoding mediators of cytokine/growth factor signalling, such as *Nr4a1, Nfkbia, Fosb* and *Junb* (*Ly6c1*+|*Hmox1*+*C1q*−) (Fig. 4a). The two *Ly6c1*+|*Hmox1*− clusters, termed *Ly6c1*+|*Hmox1*−*Ccl5*[lo] and *Ly6c1*+|*Hmox1*−*Ccl5*[hi], closely resembled each other (Fig. 1d and Supplementary Fig. 1c). Flow cytometric analysis validated the existence of *Ly6c1*+ FC expressing high or low levels of HMOX1 protein (Fig. 4d). The fractional abundance of HMOX1[hi] FC was consistent with the frequency of the *Ly6c1*+|*Hmox1*+*C1q*+ subcluster expressing the highest level of *Hmox1* mRNA (Fig. 4a, e).

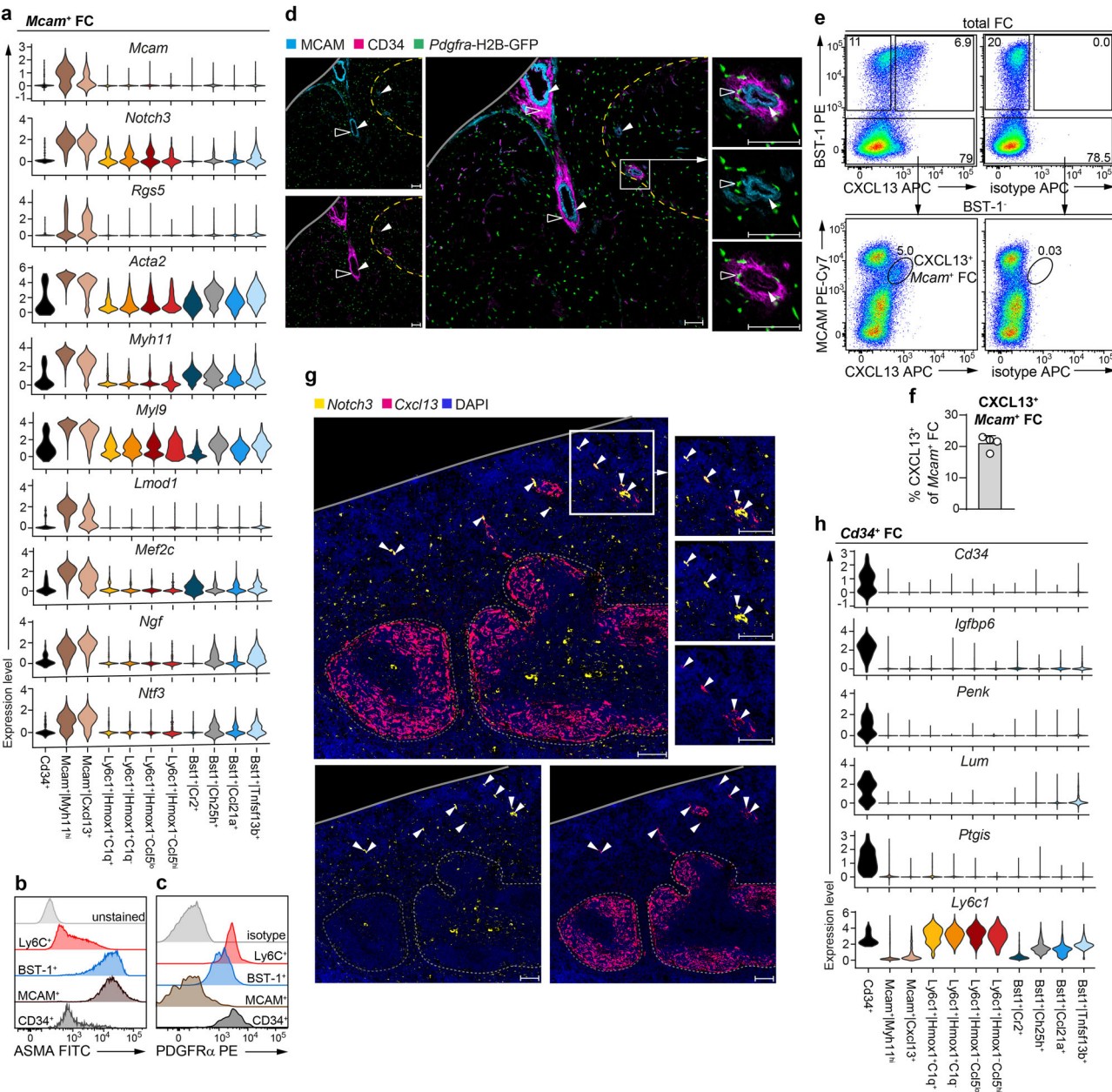

**Fig. 3 Mcam⁺ FC and Cd34⁺ FC are distinct components of the splenic perivascular niche. a** Violin plots showing expression of marker genes that segregate *Mcam*⁺ FC (scaled total UMI counts per cell) across individual clusters. **b, c** Flow cytometric analysis of **b** ASMA or **c** PDGFRα expression by the indicated FC subsets. Representative stains from 2 independent experiments with a single biological replicate per experiment using pooled cell preparations from 2 spleens/replicate. **d** Confocal microscopy analysis of MCAM (light blue), CD34 (pink) on splenic sections from mice expressing H2B-GFP fusion gene (i.e. nucleus-confined GFP) from the endogenous *Pdgfra* locus (green). Exemplary *Mcam*⁺ FC (MCAM⁺GFP⁻) are indicated by filled arrowheads. Exemplary *Cd34*⁺ FC (CD34⁺GFP⁺) are indicated by empty arrowheads. Dashed yellow line, the white pulp/red pulp border. Solid grey line, the capsule. Scale bars are 50 μm. Representative stains from 3 spleens analysed in 2 independent experiments. **e, f** Flow cytometric validation of *Mcam*⁺|*Cxcl13*⁺ FC. **e** Representative stains. Numbers are percentage of cells in the indicated gates. **f** Percentage of CXCL13⁺ cells amongst *Mcam*⁺ FC. Data are pooled from 2 independent experiments and presented as arithmetic mean ± SD of $n = 4$ mice (depicted as symbols). **g** In situ RNA hybridization (RNAscope) for *Cxcl13* (pink) and *Notch3* (yellow). Sections were counterstained with DAPI (dark blue). Exemplary *Mcam*⁺|*Cxcl13*⁺ FC (Notch3⁺Cxcl13⁺ cells) are indicated by arrowheads. Dashed grey line, the white pulp/red pulp border. Solid grey line, the capsule. Scale bars are 100 μm. Representative stains from 2 spleens analysed in 2 independent experiments. **h** Violin plots showing expression of marker genes that segregate *Cd34*⁺ FC (scaled total UMI counts per cell) across individual clusters. **a, h** Violin plots were generated using default settings of the function VlnPlot() of the R package Seurat.

**Ly6c1⁺ FC are enriched for antiviral gene expression at steady state**. Next, to further explore functions of splenic FC, we averaged single-cell transcriptomes of the main FC fractions, and applied gene ontology analysis of biological processes based on the top 300 cell type-specific DEGs (log₂(fold change) > 0.25, *p*-val < 0.05). Notably, *Ly6c1*⁺ FC showed specific

over-representation of genes involved in the inhibition of virus replication and defence response to virus infection (Fig. 5a). To assess this antiviral propensity in more detail on the level of individual FC clusters, we next analysed the expression of 104 ISGs, common to both human and mouse lymphoid and myeloid cells[28] across all eleven FC clusters. In agreement with the gene

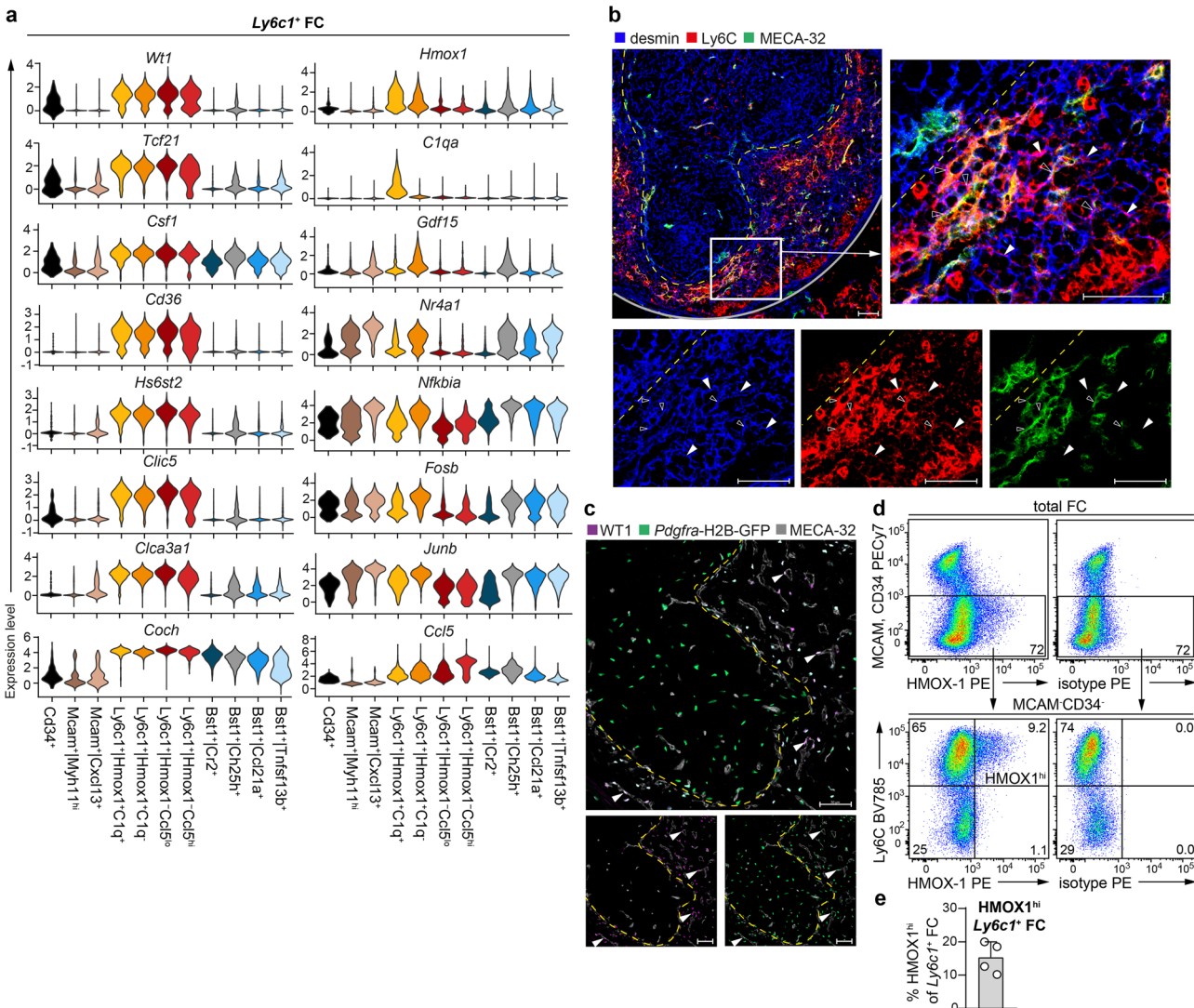

**Fig. 4 Ly6c1+ FC are localized in the splenic red pulp and the marginal zone. a** Violin plots showing expression of marker genes that segregate *Ly6c1+* FC (scaled total UMI counts per cell) across individual clusters. Violin plots were generated using default settings of the function VlnPlot() of the R package Seurat. **b** Confocal microscopy analysis of Ly6C (red), desmin (blue) and MECA-32 (green). Filled arrowheads point to exemplary *Ly6c1+* FC (Ly6C +desmin+) localized in-between MECA-32+ sinusoids. Empty arrowheads point to exemplary *Ly6c1+* FC (Ly6C+desmin+) which appear in direct contact with MECA-32+ sinusoids. Dashed yellow line, the white pulp/red pulp border. Grey line, the capsule. Scale bars are 50 μm. Representative stains from 2 spleens analysed in 2 independent experiments. **c** Confocal microscopy analysis of WT1 (purple) and MECA-32 (grey) on splenic sections from mice expressing H2B-GFP fusion gene (i.e. nucleus-confined GFP) from the endogenous *Pdgfra* locus (green). Arrowheads point to exemplary *Ly6c1+* FC (WT1+GFP+) in direct contact with MECA-32+ sinusoids. Dashed yellow line, the white pulp/red pulp border. Scale bars are 50 μm. Representative stains from 3 spleens analysed in 3 independent experiments. **d, e** Flow cytometric validation of *Ly6c1+* FC expressing high levels of HMOX-1 (HMOX-1hi). **d** Representative stains. HMOX1hi *Ly6c1+* FC are indicated in the plot. Numbers shown are percentage of cells in the indicated gates. **e** Percentage of HMOX1hi cells amongst *Ly6c1+* FC. Data are pooled from 2 independent experiments and presented as arithmetic mean ± SD of n = 4 mice (depicted as symbols).

ontology analysis, the expression levels of a major proportion of ISGs were elevated in all four *Ly6c1+* FC clusters compared to other FC clusters (Fig. 5b). ISGs with upregulated expression in *Ly6c1+* FC encompassed a marked variety of genes with important roles in antiviral defence, including *Eif2ak2* (Protein kinase R), *Mlkl, Ifi44*, members of the *Oas, Gbp, Ifit* and *Ifitm* families as well as mediators of viral sensing, such as *Irf7, Zbp1* (DNA sensor, DAI), *Dhx58, Ddx60* and *Ifih1* (RNA sensor, Mda5)[11] (Fig. 5b). To independently confirm as well as further quantify the observed phenomenon, we next analysed the bulk transcriptomes of *Ly6c1+* FC, *Bst1+* FC and splenic endothelial cells (EC) by RNA-seq (Fig. 5c). Consistent with the results obtained on the single-cell level, geneset enrichment analysis demonstrated

significant enrichment for ISGs in *Ly6c1+* FC compared to *Bst1+* FC (Fig. 5d). Accordingly, *Ly6c1+* FC significantly overexpressed 46 ISGs (log2(fold change) > 0.8, p-val < 0.05) while only 11 ISGs were overexpressed by *Bst1+* FC (Fig. 5e). It deserves to be mentioned that ISGs upregulated in *Ly6c1+* FC over *Bst1+* FC matched between the RNA-seq and scRNA-seq data sets (Fig. 5b, ISGs upregulated in *Ly6c1+* FC based on the RNA-seq analysis are indicated in red). Importantly, *Ly6c1+* FC were enriched for ISG expression not only relative to other FC, but also relative to splenic EC, indicating that the observed antiviral profile of *Ly6c1+* FC does not simply reflect their closeness to the blood stream but rather is regulated by organ-specific extrinsic and/or intrinsic mechanisms (Fig. 5f). In sum, these results suggest that

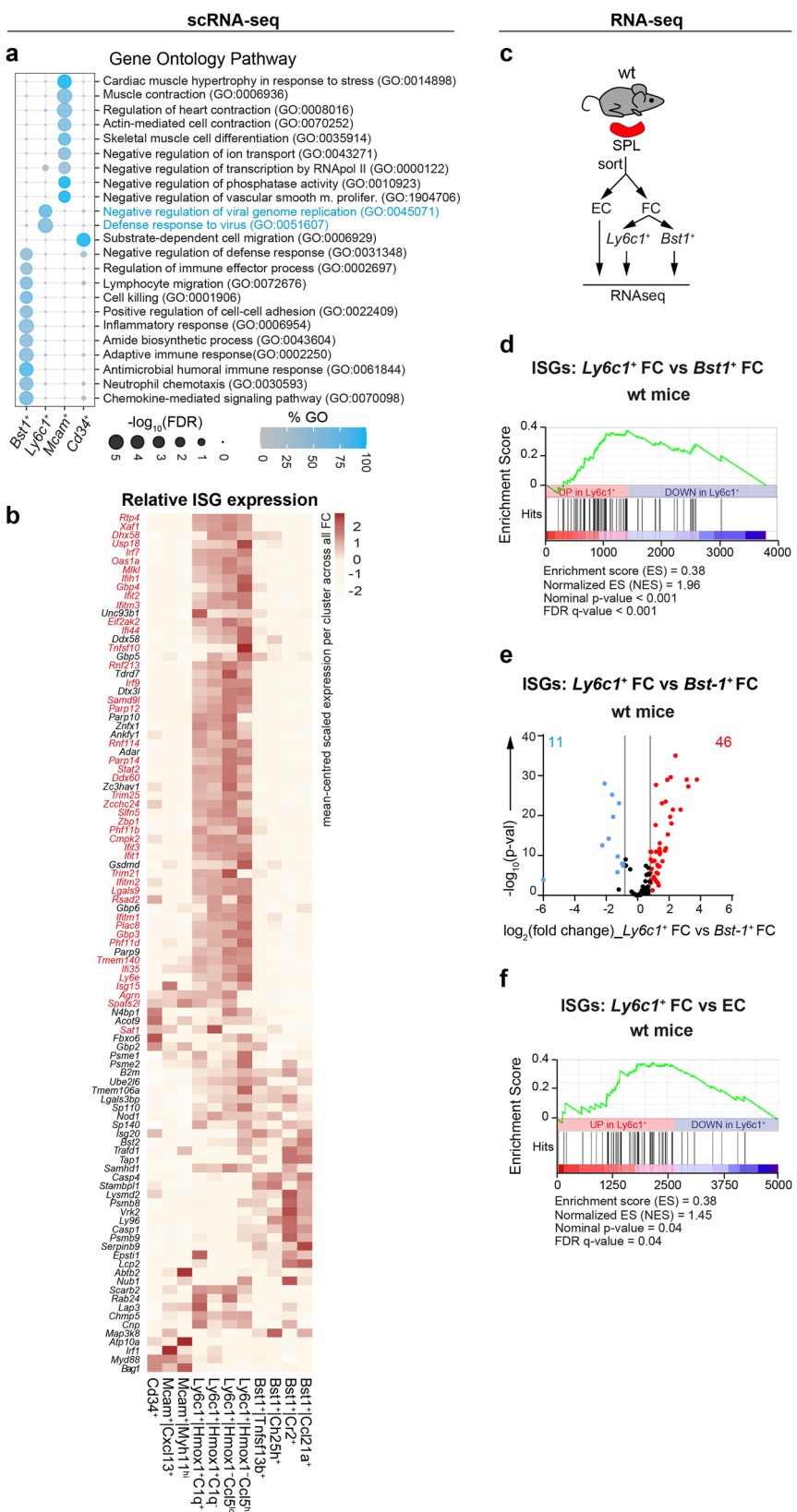

*Ly6c1$^+$* FC may show enhanced antiviral activity under homeostatic conditions.

**Antiviral signature of *Ly6c1$^+$* FC is dependent on *Stat1*.** Next, we investigated whether the enhanced antiviral gene signature of

*Ly6c1$^+$* FC is regulated by tonic IFN signalling. Firstly, we examined the impact of *Stat1*, which is required to sustain tonic IFN-dependent ISG expression[29], on the relative difference in ISG expression between *Ly6c1$^+$* FC and *Bst1$^+$* FC. To this end, *Ly6c1$^+$* FC and *Bst1$^+$* FC were sorted from *Stat1*$^{KO}$ mice and analysed by RNA-seq (Fig. 6a). To ensure correct subset

**Fig. 5 $Ly6c1^+$ FC are enriched for antiviral gene expression. a** GO analysis of biological processes for top 300 upregulated DEGs in scRNA-seq dataset across all cells per indicated FC subset. Shown are significantly enriched GO identifiers. GO identifiers significantly enriched in $Ly6c1^+$ FC that relate to antiviral defence are highlighted in blue. **b** Heatmap showing mean-centred scaled expression of individual ISGs per cluster across all FC based on scRNA-seq analysis. Indicated in red are ISGs with upregulated expression ($\log_2$(fold change) > 0.8, adj. p-val < 0.05) in bulk $Ly6c1^+$ FC versus $Bst1^+$ FC based on RNA-seq analysis of these subsets isolated from the spleens of wild-type (wt) mice (as explained in **c–f**). **c–f** RNA-seq analysis of $Ly6c1^+$ FC, $Bst1^+$ FC and EC isolated from the spleens of wt mice. Data are from 3 biological replicates, with respective subsets sorted from pooled cell preparations of 2–4 mice/ replicate. **d** Geneset enrichment analysis for ISGs performed on DEGs ($\log_2$(fold change) > 0.8, adj. p-val < 0.05) from the comparison between $Ly6c1^+$ FC versus $Bst1^+$ FC. **e** Volcano plot comparing fold change of ISG **e**xpression between $Ly6c1^+$ FC versus $Bst1^+$ FC. ISGs with upregulated expression in $Ly6c1^+$ FC or $Bst1^+$ FC ($\log_2$(fold change) > 0.8, adj. p-val < 0.05) are highlighted respectively in red and blue. **f** Geneset enrichment analysis for ISGs performed on DEGs ($\log_2$(fold change) > 0.8, adj. p-val < 0.05) from the comparison between $Ly6c1^+$ FC versus EC.

identification (note $Ly6c1$ is a type I IFN-inducible gene[30] expressed by $Ly6c1^+$ FC of $Stat1^{KO}$ mice at a modestly reduced level), $Ly6c1^+$ FC were sorted as MCAM−CD34−BST-1−PDGFRα+ FC (Supplementary Fig. 2a). Notably, geneset enrichment analysis performed on bulk transcriptomes of $Ly6c1^+$ FC and $Bst1^+$ FC from $Stat1^{KO}$ mice revealed that in the absence of $Stat1$, $Ly6c1^+$ FC were no longer enriched for ISG expression compared to $Bst1^+$ FC (Fig. 6b). Accordingly, loss of $Stat1$ resulted in overall lower fold change difference in ISG expression between $Ly6c1^+$ FC versus $Bst1^+$ FC (Fig. 6c). Specifically, $Stat1$ was responsible for the overexpression of 26 of 46 genes that constituted the ISG signature of $Ly6c1^+$ FC but affected none of the 11 ISGs that were overexpressed by $Bst1^+$ FC (Supplementary Table 1). $Stat1$-dependent enrichment for ISG expression in $Ly6c1^+$ FC was corroborated using an independent ISG set collated for primary fibroblasts (extracted from the Interferome database v2.0[31] using the following search criteria: max. 6 h post stimulation with IFNβ, fold change > 2.5; p-val < 0.05) (Supplementary Fig. 2b). Finally, in the absence of $Stat1$, $Ly6c1^+$ FC were also no longer enriched for the expression of genes involved in defence response to virus as revealed by gene ontology analysis (Fig. 6d). Thus, the selectively augmented antiviral gene signature of $Ly6c1^+$ FC is $Stat1$-dependent. These results were consistent with a role for tonic IFN signalling in sustaining the transcriptionally enhanced antiviral profile of $Ly6c1^+$ FC. To corroborate this, we assessed the impact of IFNAR loss on the expression of select ISGs, which are overexpressed in $Ly6c1^+$ FC in wt mice. To this end, $Ly6c1^+$ FC and $Bst1^+$ FC were sorted from wt and $Ifnar1^{KO}$ mice and analysed by RT-qPCR (Fig. 6e). Indeed, loss of IFNAR expression equalized expression levels of all tested ISGs between $Ly6c1^+$ FC and $Bst1^+$ FC (Fig. 6f). Furthermore, $Ifnar1$ and $Stat1$ similarly affected the expression of ISGs involved in IFN signalling (Supplementary Fig. 2c, d). Collectively, the presented evidence indicates that the augmented antiviral signature of $Ly6c1^+$ FC is induced and/or sustained by tonic type I IFN signalling.

**Antiviral gene expression by splenic FC following immune stimulation in vivo.** Next, we addressed the question whether in the presence of acutely produced type I IFNs, $Ly6c1^+$ FC would also express ISGs on a relatively higher level. To study the response of splenic FC to type I IFNs disseminating from a peripheral site, we analysed ISG expression in splenic FC subsets and EC following a single dose of IFN-β delivered subcutaneously (Fig. 7a). Two hours after stimulation, which is sufficient to achieve maximal ISG response in cells residing in the white pulp of the spleen[28], indicated subsets were sorted and analysed by RT-qPCR. As shown in Fig. 7a, $Ly6c1^+$ FC of IFN-treated mice attained the highest expression of all tested ISGs. $Ly6c1^+$ FC expressed these ISGs at a higher level independently of the strength of the stimulus, as underscored by dose-response analysis (Fig. 7b). Next, we studied antiviral gene expression by splenic FC subsets in response to the infection with mouse

cytomegalovirus, MCMV in vivo. Based on histological analysis, it has previously been indicated that MCMV administered intraperitoneally infects and undergoes the first round of replication in stromal cells located in the splenic marginal zone and the red pulp[32], a localization corresponding to $Ly6c1^+$ FC. Flow cytometric analysis of splenic stromal cells 12 h post intraperitoneal infection with $10^6$ PFU of MCMV[GFP], demonstrated that $Ly6c1^+$ FC were infected at the frequency of ca 3% whereas infection rates of other analysed FC subsets, $Bst1^+$ FC and $Mcam^+$ FC, were substantially lower (Fig. 7c). To obtain a snapshot of antiviral gene expression across all FC subsets in MCMV-infected mice, we performed scRNA-seq analysis of splenic FC 24 h post intraperitoneal infection with $10^6$ PFU of MCMV (Fig. 7d). In keeping with the flow cytometric analysis (Fig. 7c), MCMV-infected cells (3.1%) resided in a distinct cluster that expressed markers of cellular stress and clustered closely with $Ly6c1^+$ FC (Supplementary Fig. 3a). Virus-infected cells were removed from subsequent analysis as we aimed to assess the bystander immune response of FC and not their cell-intrinsic response to the virus. After quality control and exclusion of mesothelial, endothelial, and virus-infected cells, we retained for further analysis 25,419 FC with 2815 median detected genes per cell (Supplementary Fig. 3b). The identity of the eleven FC clusters described in the steady-state condition was preserved among splenic FC from infected mice (Fig. 7e, Supplementary Fig. 3c). The relative abundance of individual clusters was similar between both conditions, except for a reciprocal shift between the two most similar $Ly6c1^+$ clusters, $Ly6c1^+|Hmox1^-Ccl5^{lo}$ and $Ly6c1^+|Hmox1^-Ccl5^{hi}$ (these cells were not proliferating), suggesting they may represent alternative activation states of the same subset (Figs. 1b and 7e). Based on the comparison of cZ-scores for ISG expression calculated for individual subsets in the untreated versus infected condition, all FC clusters upregulated ISGs following MCMV infection, with the highest cZ-scores noted for $Ly6c1^+$ FC and for a fraction of $Cd34^+$ FC (Fig. 7f). A more detailed evaluation on the level of individual ISGs revealed that even though some $Cd34^+$ FC reached a similar ISG cZ-score as $Ly6c1^+$ FC, $Ly6c1^+$ FC were the only FC subset uniquely overexpressing a sizeable array of ISGs in virus-infected mice (Fig. 7g). Notably, ISGs selectively overexpressed by $Ly6c1^+$ FC in virus-infected mice overlapped with ISGs overexpressed by these cells in the steady state (Fig. 7g). Further importantly, $Ly6c1^+$ FC in virus-infected mice were, like in the steady state, selectively enriched for antiviral gene expression, as underscored by gene ontology analysis (Fig. 7h). Finally, we complemented the scRNA-seq analysis performed at 24 h post infection with a time-resolved profile of ISG expression by splenic FC in virus-infected mice. To this end, we performed a kinetic analysis of the expression of $Irf7$ and $Oas1a$, which are overexpressed in $Ly6c1^+$ FC at 24 h post MCMV infection (Fig. 7g), in $Ly6c1^+$ FC and $Bst1^+$ FC at 12, 36 and 48 h post intraperitoneal infection with $10^6$ PFU of MCMV by RT-qPCR. The ISG response of splenic FC was biphasic, matching the biphasic kinetic of type I IFN

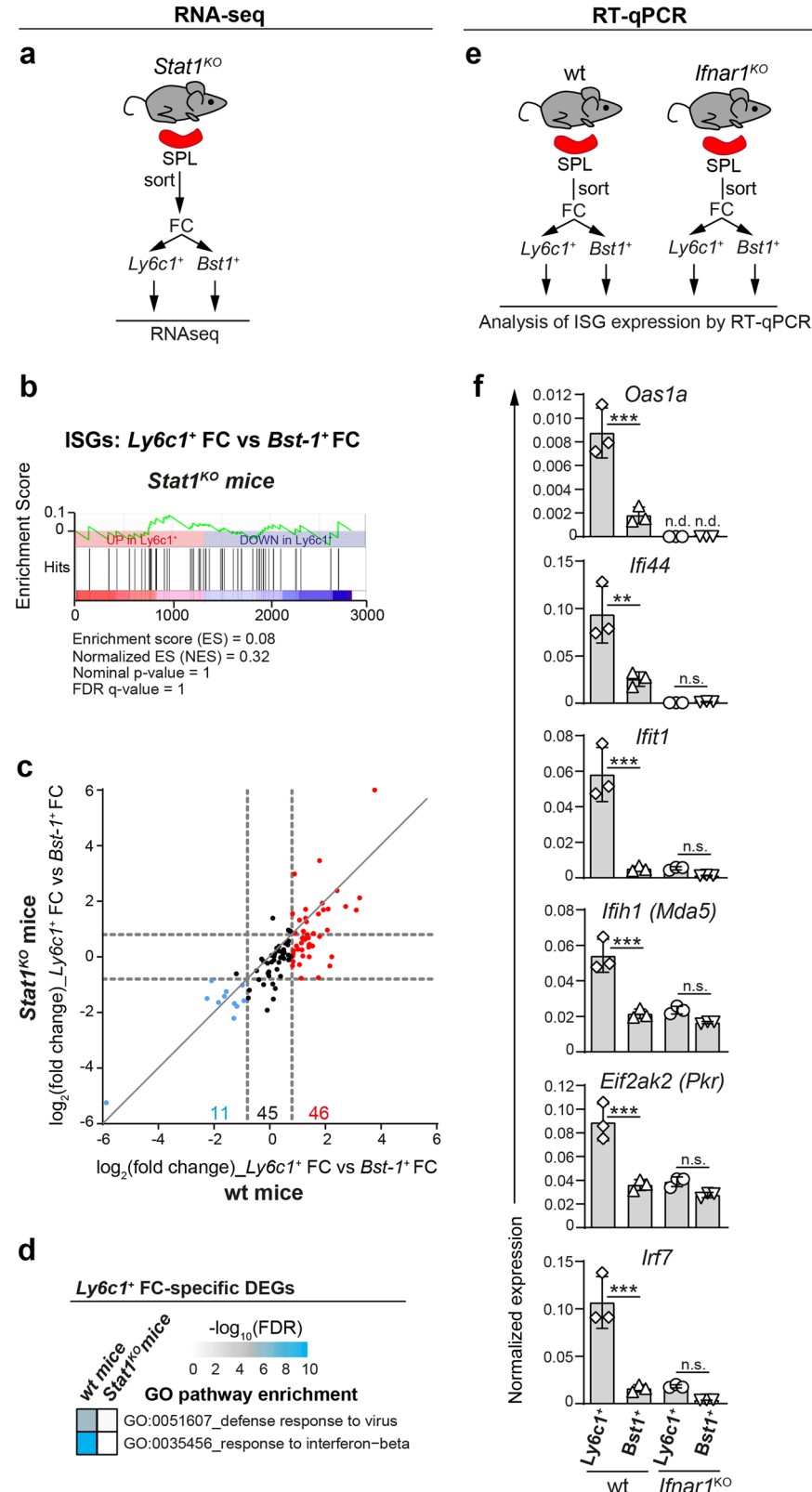

production upon MCMV infection, which peaks in the first 24 h and then again at 48 h post infection[33]. Importantly, $Ly6c1^+$ FC expressed the tested ISGs at a higher level compared to $Bst1^+$ FC, with highest differences at 12 h and at 48 h (Fig. 7i). In sum, the transcriptionally augmented antiviral profile of $Ly6c1^+$ FC across various immune conditions, suggests a specialized innate immune function of these cells. The putative importance of $Ly6c1^+$ FC in innate antiviral defence remains to be directly addressed by future studies using conditional knockout models.

## Conclusions

Here, we performed single-cell transcriptional analysis of FC from adult mouse spleen, identifying eleven transcriptionally

**Fig. 6 Antiviral signature of *Ly6c1*$^+$ FC is dependent on *Stat1*. a–d** RNA-seq analysis of *Ly6c1*$^+$ FC and *Bst1*$^+$ FC purified from the spleens of *Stat1*$^{KO}$ mice. Data are from 3 biological replicates, with respective subsets sorted from pooled cell preparations of 2 mice/replicate. **b** Geneset enrichment analysis for ISGs performed on DEGs (log$_2$(fold change) > 0.8, adj. *p*-val < 0.05) from the comparison between *Ly6c1*$^+$ FC versus *Bst1*$^+$ FC in *Stat1*$^{KO}$ mice. **c** Scatter plot depicting fold change in the expression of individual ISGs between *Ly6c1*$^+$ FC versus *Bst1*$^+$ FC in (*x*-axis) wt and (*y*-axis) *Stat1*$^{KO}$ mice. ISGs with upregulated expression in *Ly6c1*$^+$ FC or *Bst1*$^+$ FC (log$_2$(fold change) > 0.8, adj. *p*-val < 0.05) in the wt condition are highlighted respectively in red and blue. **d** GO analysis of biological processes for top 1188 DEGs with upregulated expression in *Ly6c1*$^+$ FC versus *Bst1*$^+$ FC (log$_2$(fold change) > 0.8, adj. *p*-val < 0.05) in the wt condition or in *Stat1*$^{KO}$ mice. Shown are significantly enriched GO identifiers that relate to antiviral defence. **e, f** RT-qPCR analysis of ISG expression in *Ly6c1*$^+$ FC and *Bst1*$^+$ FC isolated from the spleens of wt and *Ifnar1*$^{KO}$ mice. **f** Data are pooled from 2 independent experiments and presented as arithmetic mean ± SD of *n* = 3 biological replicates (depicted as symbols) using cells sorted from pooled cell preparations of 2 mice/replicate. Statistical significance was calculated using one-way ANOVA with Sidak's multiple comparison test. **p* < 0.01; ***p* < 0.001; n.s. denotes *p* > 0.05; n.d. denotes not detected.

---

distinct FC clusters distributed across four main cell identities, such as (i) *Bst1*$^+$ white pulp fibroblasts comprising four clusters; (ii) *Ly6c1*$^+$ red pulp fibroblasts comprising four clusters; (iii) *Mcam*$^+$ mural cells with pericyte/VSMC identity comprising two clusters and (iv) a single cluster of *Cd34*$^+$ adventitial fibroblasts. In addition to confirming the subdivision of white pulp FC into TRC, MRC and FDC, our analysis resolved a fourth, functionally specialized *Dpt*$^+$*Tnfsf13b*$^+$ FC subset localized in the splenic T-cell zone. Detailed characterization of the FC components of the splenic perivascular niche discerned a subset of CXCL13-expressing *Mcam*$^+$ pericyte/VSMC present around select vessels in the splenic red pulp. We further uncover that *Ly6c1*$^+$ red pulp FC overexpress a transcriptional signature composed of genes involved in defence response to viral infection, underpinned by overexpression of ISGs, under homeostatic conditions and upon virus infection in vivo; suggesting a potential specialized role for splenic red pulp stroma in antiviral defence. In sum, our characterization of splenic FC at single-cell level provides insight into the functional diversity of splenic stromal cells.

## Methods

**Mice**. Unless otherwise indicated, experiments were performed with age- and sex-matched 2–5-m-old mice on C57BL/6 background. C57BL/6JrJ mice were purchased from Janvier Labs. *Ifnar1*$^{KO}$[34], *Stat1*$^{KO}$[35] and *Pdgfra*$^{H2B-GFP}$[36] mice were bred and maintained under specific pathogen free (SPF) conditions according to Federation of European Laboratory Animal Science Associations (FELASA) guidelines, respectively, at central animal facility of HZI Braunschweig, Germany; University of Veterinary Medicine Vienna, Austria and San Raffaele Scientific Institute, Milan, Italy. Animal procedures were approved by the responsible state office (Lower Saxony State Office of Consumer Protection and Food Safety).

**Viral infections**. Infection of mice was performed by intraperitoneal administration of 10$^6$ PFU of either BAC-derived MCMV (clone pSM3fr-MCK-2fl 3.3[37]) or BAC-derived MCMV-GFP-P2A-ie1/3, MCMV$^{GFP}$. MCMV-GFP-P2A-ie1/3 was generated by *en passant* BAC mutagenesis[38] using the BAC pSM3fr-MCK-2fl 3.3[37]. Briefly, the gene encoding GFP was fused with the MCMV gene encoding immediate early transcripts 1 and 3 (m122/123)). The P2A peptide-encoding sequence was inserted after GFP removing its stop codon generating a bicistronic ORF. Cloning design was carried out using SnapGene software (GSL Biotech, USA). The recombinant BAC was transfected into NIH3T3 cells using FuGene HD (Promega, USA) and reconstituted viral particles were passaged five times before generating a stock from infected M2-10B4 mouse fibroblastic cells (ATCC CRL-1972).

**Cell Isolation**. Spleens were digested with collagenase P (0.4 mg/ml), dispase II (2 mg/ml) and DNase I (50 µg/ml) in RPMI-1640 supplemented with 1 mM sodium pyruvate, 100 U/ml penicillin, 100 U/ml streptomycin, 10 mM HEPES and 5% fetal bovine serum (FBS). Spleens were pre-incubated with the above digestion solution injected into the organ using a 26 G needle for 5 min at RT, then minced using scissors and digested for 30 min at 37 °C. Enzymatic treatment was repeated for additional 20 min followed by incubation with 5 mM EDTA at RT for 5 min. The resulting cell suspensions were passed through a 70 µm filter, followed by immunomagnetic depletion of CD45$^+$ cells using MACS (Miltenyi Biotech). Briefly, cells from one spleen were incubated with 60 µl anti-CD45 microbeads in 600 µl of PBS containing 2 mM EDTA and 2% FBS for 20 min on ice, washed, and depleted on LS columns according to manufacturer's instructions (Miltenyi Biotech).

**Flow cytometry and cell sorting**. Flow cytometry was performed with antibodies listed in Supplementary Table 2. Dead cells (identified using 7-AAD Viability Staining Solution or Zombie NIR Fixable Viability Kit; both from BioLegend) and cell aggregates (identified on FSC-A versus FSC-H scatter plots) were excluded from all analyses. For intracellular staining, surface-labelled cell suspensions were fixed using eBioscience Foxp3/Transcription Factor Staining Buffer Set or eBioscience IC Fixation Buffer (both from Thermo Fisher). HMOX-1 expressing cells were detected with anti-HMOX-1 antibody coupled to PE using the Lightning-Link conjugation kit (abcam). Data acquisition was performed on an Aria-II SORP, ARIA-Fusion or LSR-Fortessa (BD Biosciences) and analysed using FlowJo software (BD Biosciences). Sorting was performed on an Aria-II SORP or ARIA-Fusion (BD Biosciences).

**Immunofluorescence staining of splenic sections**. Spleens were fixed in 4% (wt/vol) paraformaldehyde (PFA) for 10 min, saturated overnight in 30% (wt/vol) sucrose at 4 °C, and embedded in Tissue-Tek optimum cutting temperature compound (Sakura) followed by freezing in −80 °C. For detection of Ly6C and BST-1, sections were fixed prior to staining with ice-cold acetone for 10 min. Sections (7 µm) were blocked with 10% (vol/vol) goat serum. Endogenous peroxidase and biotin activities were quenched respectively with 3 % (vol/vol) hydrogen peroxide solution and Endogenous Biotin-Blocking Kit (Thermo Fisher). Antibodies (listed in Supplementary Table 2) were diluted in PBS containing 0.05% (vol/vol) Tween-20 and 2% (vol/vol) goat serum. Primary biotinylated antibodies were visualized with HRP-conjugated streptavidin followed by TSA Plus Cyanine 3 or Cyanine 5 System (Akoya). BST-1 expressing cells were detected with anti-BST-1 antibody coupled to Alexa647 using the Lightning-Link conjugation kit (abcam). Images were acquired with ZEISS LSM 980 confocal microscope and analysed using ZEN software (both from Carl Zeiss MicroImaging).

**In situ RNA hybridization (RNAscope)**. In situ RNA hybridization was performed using the RNAscope Multiplex Fluorescent Detection Kit v2 (Advanced Cell Diagnostics) according to manufacturer's instructions. The following target probes were used: Mm-Dpt (Cat. #561511-C3), Mm-Tnfsf13b (Cat. #414891), Mm-Notch3 (Cat. #425171), Mm-Cxcl13 (Cat. #406311-C2). In brief, spleens were fixed in 10% formalin for 24 h at RT and embedded in paraffin. 3 µm-thick sections were baked in an oven at 60 °C for 1 h, then deparaffinized and dehydrated. Following rehydration, endogenous peroxidase activity was quenched with hydrogen peroxide for 10 min at RT. Target retrieval was carried out in a steamer (Braun, Type 3216) for 15 min at >98 °C. Protease treatment was performed with Protease Plus (Advanced Cell Diagnostics) for 20 min at 40 °C. Target probes were allowed to hybridize for 2 h at 40 °C, followed by signal amplification according to the ACD protocol. After the final amplification step, signal was detected with Opal520, Opal570 or Opal650-conjugated tyramide (Perkin Elmer). Sections incubated with a negative control probe (*DapB*) were analysed in parallel and a mix of positive control probes (*Polr2a*, *Ppib*, *Ubc*) was utilized to confirm RNA integrity. Images were acquired with an Olympus VS120 slide scanner fluorescence microscope using the VS-ASW-FL software (Olympus).

**RT-qPCR**. Total RNA was extracted using RNeasy Plus Micro Kit (Qiagen), reverse transcribed with SuperScript IV and a 1:1 mixture of oligo-dT and random oligonucleotide hexamers (all from Thermo Fisher). Quantitative PCR was performed using Forget-Me-Not EvaGreen qPCR Master Mix (Biotium) in a LightCycler 480 Instrument II (Roche). Relative gene expression was calculated using the 2$^{-\Delta CT}$ method with normalization to the expression of *Gapdh*. Primers (listed in Supplementary Table 3) were validated for the use of the 2$^{-\Delta CT}$ method by determining the efficiency of each primer pair in the corresponding expression range.

**Single-cell RNA sequencing analysis**. Cells were sorted into DMEM medium containing 10% FBS and adjusted to a density of 1000 cells/µl. Chromium Controller (10x Genomics) was used for partitioning single cells into Gel Bead-In-EMulsions (GEMs) and Chromium Single Cell 3′ GEM, Library & Gel Bead Kit v3 (10x Genomics) for reverse transcription, cDNA amplification and library

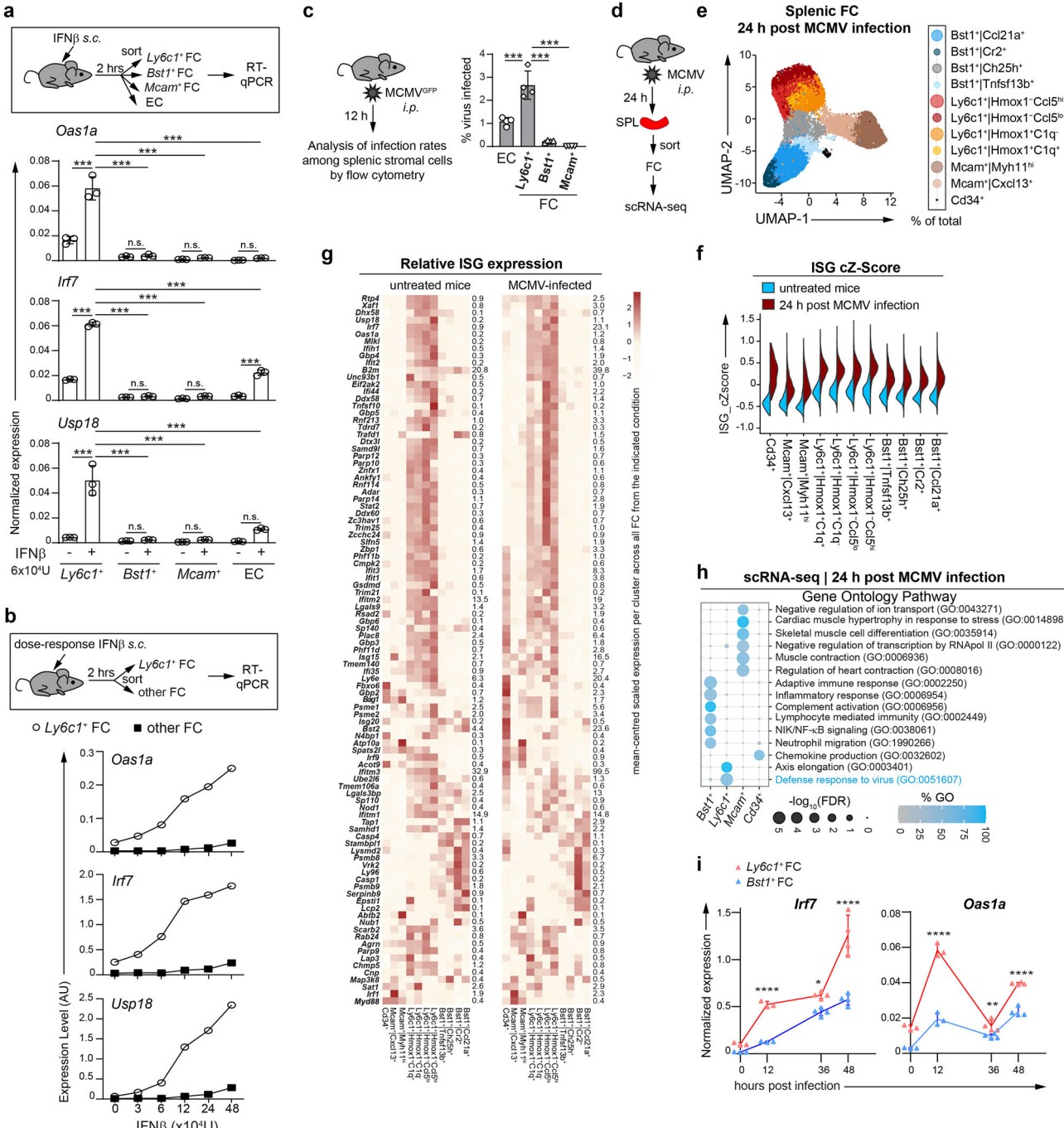

**Fig. 7 Antiviral gene expression by splenic FC following immune stimulation in vivo. a, b** RT-qPCR analysis of ISG expression in indicated splenic subsets following a 2 h stimulation of mice with the indicated dose of IFNβ *s.c.* **a** Data from singular in vivo experiment is presented as arithmetic mean ± SD of *n* = 3 biological replicates (depicted as symbols) using cells sorted from pooled cell preparations of 2 mice/replicate. **b** Data from singular dose-response experiment with one mouse per each indicated IFNβ dose. Symbols depict ISG expression in indicated subsets sorted from a single mouse per IFNβ dose. **c** Percentage of GFP⁺ cells among indicated splenic subsets 12 h post infection of mice with 10⁶ PFU of MCMV^GFP *i.p.* Data from one representative experiment of 2 independent experiments is presented as arithmetic mean ± SD of *n* = 4 mice (depicted as symbols). **d–h** scRNA-seq analysis of splenic FC 24 h post infection of mice with 10⁶ PFU of MCMV *i.p.* Data pooled from 2 independent experiments using cells sorted from pooled splenic preparations of 3 mice. **e** UMAP embedding of splenic FC. **f** cZ-score of ISG expression per cell across all FC clusters in the untreated or MCMV-infected condition. **g** Heatmaps showing mean-centred scaled expression of individual ISGs per cluster across FC in the untreated or MCMV-infected condition. Numbers indicate the average expression level of indicated ISGs across all FC in the indicated condition. **h** GO analysis of biological processes for top 300 upregulated DEGs across all cells per indicated FC subset in the infected condition. Shown are significantly enriched GO identifiers. GO identifiers significantly enriched in *Ly6c1*⁺ FC that relate to antiviral defence are highlighted in blue. **i** Kinetic assessment of ISG expression in *Ly6c1*⁺ FC and *Bst1*⁺ FC following infection of mice with 10⁶ PFU of MCMV *i.p.* determined by RT-qPCR. Data from singular in vivo experiment is presented as arithmetic mean ± SD of *n* = 3 (0, 12 h) and *n* = 4 (36, 48 h) mice (depicted as symbols). **a, c, i** Statistical significance was calculated using one-way ANOVA with Sidak's multiple comparison test. *p < 0.05; *p < 0.01; ***p < 0.001; ****p < 0.0001; n.s. denotes p > 0.05.

construction (all performed according to manufacturer's instructions). Libraries were sequenced on a NovaSeq 6000 sequencer (Illumina) using NovaSeq 6000 S1 Reagent Kit (100 cycles, 28 bp read 1, 89 bp read 2) and attained approximately 34,100 reads per single cell across all replicates. The GEO accession number for the scRNA-seq data reported in this paper is GSE156162. Raw base call (BCL) files were demultiplexed to generate Fastq files using the cellranger *mkfastq* pipeline within Cell Ranger 3.0.3 (10x Genomics). Whole transcriptome Fastq files were processed using the standard cellranger pipeline within Cell Ranger 3.0.3 (10x Genomics) against the merged genomes of GRCm38 (Ensembl v90) and MCMV (BAC-derived wild-type Smith strain, Accession Identifier: NC_004065). The expected cell parameter was set to 10,000 for all libraries. Briefly, cellranger performs alignment, filtering, barcode counting, and UMI counting to obtain read count matrices (RCM). In total four RCMs were obtained, with two independent experiments performed for both steady state and virally infected condition. RCMs were further processed via R (version 3.6.2) using *Seurat* (version 3.1.5)[39] and *uwot* (version 0.1.8)[40]. Cells with at least 1000 detected genes and <7.5% mitochondrial reads were retained in the analysis. Analysis was performed in a stepwise manner to successively identify and focus on splenic FC. In total, 50,401 cells with 16,426 detected genes were processed. Genes were included if they were expressed in at least 1% of the cells. Initially, the four RCMs were merged using *FindIntegration Anchors (list(experimental_batches), anchor. features = 1000, dims = 1:15, k. filter = 200, k. anchor = 10)* and *Integrate Data()*. Data was scaled and regressed against the variables: percent mitochondrial UMIs, percent ribosomal protein UMIs, percent viral UMIs and the total number of UMIs per cell. PCAs were computed using default settings. Uniform Manifold Approximation and Projection (UMAP) dimensional reduction via *RunUMAP()* and *FindNeighbors()* were performed using the first 15 PCA dimensions as input features. *Find Clusters()* was computed at resolution 0.4. The respective RCMs are deposited under the GEO accession number GSE156162. $Bst1^+$ FC, $Ly6c1^+$ FC, $Mcam^+$ FC and $Cd34^+$ FC were identified based on the expression of the respective cell type markers and all cells residing within these clusters were re-embedded as detailed above with the following parameters: *anchor. features = 1000, dims = 1:17, k. filter = 200, k.anchor = 10, resolution = 0.45* and the first 17 dimensions used as input features. Merged data was visualized using the Seurat intrinsic functions *VlnPlot(), FeaturePlot(), DotPlot(), DimPlot()*. Differentially expressed genes per cluster were identified using *FindAllMarkers()* or *FindMarkers()*. From two independent experiments performed for both steady state and virally infected condition, $Bst1^+$ FC, $Ly6c1^+$ FC, $Mcam^+$ FC and $Cd34^+$ FC collectively amounted to 47,039 cells with 14,306 detected genes were processed. Cells with a proportion of >1% viral UMIs were excluded. When analysing distinct subsets, all cells residing within the respective cluster were re-embedded as detailed above with the following parameters: *anchor. features = 500, dims = 1:4, k. filter = 200, k. anchor = 10, resolution = 0.1* and the first 4 dimensions used as input features. Differentially expressed genes per cluster were identified using *Find All Markers()*. Cumulative Z-scores were calculated based on the scaled expression per gene across all clusters for the given comparison or all cells and summed for the defined gene signatures. Gene ontology analysis was performed for the indicated sets of differentially expressed genes using *topGO*[41]. Pie-chart, bubble-plot and bar graph visualizations were carried out with *ggplot2*.

**RNA sequencing analysis**. Cells were sorted into RLT-Plus lysis buffer (Qiagen) containing 10 µl/ml β-mercaptoethanol. Total RNA was extracted using RNeasy Plus Micro kit (Qiagen) according to manufacturer's instructions. Quality and integrity of total RNA was controlled using 5200 Fragment Analyzer System (Agilent Technologies). For the experiment comparing $Ly6c1^+$ FC, $Bst1^+$ FC and EC from wt mice, cDNA conversion was carried out using SMART-Seq v4 Ultra Low Input RNA Kit (Takara Clontech Laboratories) according to manufacturer's instructions, followed by library generation using Nextera XT DNA Library Prep Kit (Illumina). Libraries were then sequenced on a HiSeq2500 sequencer (Illumina) using paired-end run (read 1–50 bp, read 2–30 bp) with an average of $5 \times 10^7$ reads per RNA sample. For the experiment comparing FC subsets from $Stat1^{KO}$ mice, the RNA sequencing libraries were generated using NEB Next Single Cell/Low Input RNA Library Prep Kit for Illumina (NEB) according to manufacturer's protocol and sequenced on a NovaSeq 6000 sequencer (Illumina) using NovaSeq 6000 S1 Reagent Kit (100 cycles, paired-end run $2 \times 50$ bp) with an average of $5 \times 10^7$ reads per RNA sample. The GEO accession number for all RNA-seq data reported in this paper is GSE156162. Read quality of sequenced libraries was evaluated with *FastQC*. Sequencing reads were aligned to the reference mouse genome assembly GRCm38 using STAR[42]. Reads aligned to annotated genes were quantified with *htseq-count*[43]. Raw read counts were converted to RPKM (reads per kilobase of exon length per million mapped reads) values. Protein-coding genes with at least 20 reads in at least two replicates were included in the analysis. The calculated read counts were further processed with *DESeq2* for quantification of differential gene expression[44]. Geneset enrichment analysis was performed in the pre-ranked mode using the GSEA desktop application v4.1.0 (https://www.gsea-msigdb.org/gsea). Analysis was performed on DEGs ($\log_2$(fold change) > 0.8, *p*-val < 0.05) from the comparison between $Ly6c1^+$ FC and $Bst1^+$ FC from the wild-type or $Stat1^{KO}$ condition with fold change serving as the ranking metric. The number of permutations was set to 1000.

**Statistics and reproducibility**. No statistical methods were used to predetermine sample size. One-way ANOVA with Sidak's multiple comparison test were performed with GraphPad Prism 8 to calculate statistical significance. *P*-values <0.05 were considered significant. $*p < 0.05$; $**p < 0.01$; $***p < 0.001$; $****p < 0.0001$. Error bars denote mean ± SD. The sample size for each experiment is reported in the Figure Legends. No data were removed from statistical analysis as outliers. Results have been confirmed in at least two independent experiments.

**Reporting summary**. Further information on research design is available in the Nature Research Reporting Summary linked to this article.

## Data availability

RNA-seq and scRNA-seq data generated during this study were deposited in the NCBI Gene Expression Omnibus (GEO) database with accession number GSE156162. All source data underlying the graphs shown in the main and supplementary figures are presented in Supplementary Data 2.

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

## Acknowledgements

This study was funded by the Deutsche Forschungsgemeinschaft (DFG, German Research Foundation)—Projektnummer 158989968—SFB 900, the German Centre for Infection Research (DZIF) of the German Federal Ministry of Science and Education (BMBF) through TTU 07.834 (to L.C-S.), the European Union's Horizon 2020 research and innovation program under Grant Agreement 793858 (to K.M.S.) and the Austrian Science Fund FWF SFB F6101 und F6106 (to B.S., M.M.). We thank Inge Hollatz-Rangosch and Ayse Barut for technical assistance. We acknowledge Lothar Gröbe and Maria Höxter from HZI core flow cytometry facility as well as Robert Geffers from HZI genomics platform.

## Author contributions

Concept and study design, K.M.S.; Bioinformatical analysis, J.P., F.E. and M.B.; Investigation, K.M.S., C.W., U.R. and T.B.; Methodology, K.M.S., J.P. and F.E; Resources, L.C-S., J.H., B.D., M.M., B.S., A.B. and U.K.; Funding acquisition, L.C-S., K.M.S., M.M., B.S.; Writing of the manuscript, K.M.S. and J.P. with input from other authors; Supervision, K.M.S. and L.C-S.

## Competing interests

The authors declare no competing interests.
