## [Transparent Peer Review File · Communications Biology]

Reviewers' comments:

Reviewer #1 (Remarks to the Author):

Pezoldt et al analyze mouse spleen fibroblasts using single cell RNA sequencing, revealing distinct clusters of cells or cell subtypes. They find four clusters of cells expressing Bst1, 4 clusters that expressed Ly6c1, two Mcam+ clusters and a cluster of Cd34+ cells. Analysis of the Bst1+ cells revealed FDC-like cells that expressed the marker Cr2 and Cxcl13, and TRC-like cells that expressed Ccl21 and Ccl19. Additional clustering of the Ch25h+ cluster and an Tnfsf13b+ cluster revealed further potential heterogeneity. The authors also examined gene expression in the two subsets of pericyte cells and the small group of CD34+ cells. Finally, they examined the four clusters of closely related Ly6C-expressing cells that expressed genes associated with red pulp fibroblasts.

Closer examination of the Ly6c1+ cells revealed expression of a greater number of ISGs when compared with the other splenic fibroblast subsets. Examination of Stat1-/- mice revealed that many of these ISGs were Stat1-dependent, while increased expression of selected ISGs were also reduced in IFN α 1-/- mice, indicating a role for homeostatic interferon in this gene expression profile. Injection of IFN β into mice stimulated increased Oas1a, Irf7 and Usp18 expression in Ly6c1+ fibroblasts after 2hr. Finally, they showed that splenic fibroblast subsets responded to MCMV infection by increasing ISG expression after 12hr.

The findings presented here are descriptive, but are interesting and build on previous studies (references 5 and 6 in particular), especially by adding information about red pulp fibroblast functions. Lacking is biological insight into the role of increased ISG expression by Ly6C+ fibroblasts during infection. In the present form, the study does not show if the signature of ISG expression in red pulp fibroblasts contributes to pathogen control or immunity. These gaps and omissions will be important to address for publication:

- The four Ly6c1+ subsets appear to have very similar gene expression. Are these alternate activation states or actual subtypes? Some validation of these clusters is required to ascertain if these are distinct subsets.
- In Fig. 1f, Sca-1 is now used to identify cells instead of CD34. Gating should be done with CD34 or the rationale for changing to another marker explained clearly.
- The location of the Tnfsf13b+ subclusters needs to be demonstrated to validate if these cells localize to the T-B or B follicles as suggested.
- Figure 3b: the CD34 subset is missing from the plot that shows ASMA expression.
- Where are the Cxcl13+Mcam+ localized in the spleen? This should be demonstrated.
- The Wt1 staining in Figure 4 c and d is nice, yet the Wt1+ cells all appear to be mostly Meca-32+ endothelial cells. Do the red pulp fibroblasts express Plvap that could confound this analysis? Furthermore, endothelial cells in blood vessels and sinusoids express Ly6C. It is not clear from these images which cells are fibroblasts and which are endothelial cells, even with the nuclear Pdgfra-H2B-GFP reporter. This might be due to the close association of these two cell types, but clearer images are required to demonstrate the cell identities.
- Of the 104 ISGs that were examined, 46 were higher in Ly6c1+ cells and only 11 in Bst1+ cells. Since these ISGs were identified in lymphoid and myeloid cells, it is possible that this excludes analysis of genes that are regulated by interferon in mesenchymal cell types but not in lymphoid or myeloid cells. Can this be ruled out?
- Amongst the 46 ISG identified in Ly6c1+ cells, not all were Stat1 dependent (figure 6c). They should show how many genes were stat1-dependent, which were or were not, and if the stat1-independent genes were known to be stat1-independent. Similarly, the 11 genes in Bst1+ cells were not found to be stat1-dependent. Was this expected?
- Following MCMV infection, all cell types upregulated ISGs, apparently all with a similar fold change relative to the homeostatic condition, except for the Cd34+ cells, which appeared to show a greater overall increase in cZ-score. This should be discussed. Despite this graph, the change in expression of ISGs after MCMV infection is not clear at the level of individual genes. The authors conclude: "the ISG

signature acquired by Ly6c1+ FC upon MCMV infection was similar to the one expressed by these cells in homeostatic conditions". A depiction of this data is required such as a heatmap of the uninfected versus infection condition to demonstrate how individual ISG expression is altered by infection.

- To what degree does the time point of the analysis after MCMV infection (12hr) influence these results? This might be expected to differ in the different cell types as the infection progresses and potentially change this pattern of expression.

- Is the ISG response to MCMV stat1-dependent?

- What is the biological relevance of increased ISG expression by Ly6C+ cells during homeostasis? Does the proposed innate immune function of splenic red pulp stroma provide some degree of protection from infection that the authors can demonstrate? Such biological insight is currently missing from these observations.

Reviewer #2:

This manuscript carefully details the molecular analyses of stromal cells in the spleen. While this is not the first study to endeavour to do this, the heterogeneity and complexity of this tissue framework results in new discoveries building on previous evidence. In this study, the authors identify a number of distinguishing features of different subsets of stromal cells and go on to undertake some cellular characterisation of these cells. The authors go on to focus on the Ly6c1+ subset and investigate the origins of the ISG regulated pathway to discover that Stat1 was key to induction and maintenance of this pathway in Ly6c1+ FC.

Specific questions:

1. The authors note that Ly6C is reduced in expression in Stat1 deficient cells. How can they confirm that they then recover the same population as they might identify in wildtype mice and that the low FCs are also not part of the group that are involved? Is there an independent marker that could be used to confirm this?

2. Given that following MCMV infection all FC upregulate the antiviral gene signature, what is the contribution of these lower expressing subsets to immune protection.

3. Given that Ly6C+ cells are preferentially infected but excluded from the dataset, how does this skew the data and the interpretation.

4. Was protein expression examined in the Bst1+ FC?

Figures:

1. The number of replicates for cellular experiments has not been indicated, nor whether the data are representative/pooled, and how many animals were analysed.

2. The authors have several instances where they skip over figures in their reporting eg. p 6: (Fig. 2a and Fig. 4a), similarly in skipping figures to refer to Fig. 7.

Reviewer #3 (Remarks to the Author):

In their manuscript, Pezoldt et al. investigate the heterogeneity of splenic fibroblasts in the mouse using a variety of methods. They identify four main populations of cells, as well as a number of subclusters, and characterize their response to viral infection and stimulation with type I interferons. This is a well-performed study with experiments nicely complementing each other. However, there are some weak points especially with respect to the interpretation of results that should be addressed prior to acceptance of the manuscript for publication.

- Little evidence is presented that the clusters identified really represent meaningfully different cell types or states. How was the resolution of clustering determined? I assume that FindMarkers was used with default setting, i.e. Louvain clustering. Did the authors try a different approach such as Leiden clustering? Did the authors perform stainings to assess whether the different subpopulations/states really exist?

- Since the main populations aren't very well defined on the UMAP, it would be great to see a UMAP colored by count matrix to be able to assess the quality of integration.
- Concerning this, did the authors get similar results when analyzing the non-infected data alone, without integrating data from both experiments? What was the distribution of cell types between the different groups?
- Cells containing viral reads were excluded from the analysis. I assume this was done to investigate the reaction of bystander cells rather than the cell-intrinsic immune response. In that case, however, it would be relevant to see the distribution of viral reads across all cells. If this is not clearly bimodal, i.e. viral UMI counts are low, no clear separation is possible between infected and non-infected cells. In line with this, which subpopulations of FCs tended to be infected?
- Figure 7f does not really support the statement that ISG-upregulation is specific to the Ly6c1+ subpopulation, as Cd34+ FCs appear to show an upregulation to similar levels at least in part of the cells.
- For the qPCR experiments: the normalized expression seems to indicate that the expression of some targets is orders of magnitude lower than that of the housekeeping gene used, and varies by similar levels between different conditions. Were primer efficiencies in the corresponding expression ranges determined to confirm that ddCt can be used rather than measuring a standard curve?
- Could the authors give some details about how GSEA was performed, specifically how the genes were filtered, what ranking metric was used, and in which mode GSEA was run (preranked or standard)?

"Single-cell transcriptional profiling of splenic fibroblasts reveals subset-specific innate immune signatures in homeostasis and during viral infection"

Dear reviewers,

We thank the reviewers for the valuable comments and feedback. We addressed the comments with additional experiments and analysis in line with the reviewers' recommendation and provide a revised version addressing their concerns, as well as a point-by-point reply to reviewers' remarks. In the article file, changes are highlighted in yellow.

Reviewers' comments:

Reviewer #1 basic immunology research, mesenchymal stromal cells

Pezoldt et al analyze mouse spleen fibroblasts using single cell RNA sequencing, revealing distinct clusters of cells or cell subtypes. They find four clusters of cells expressing Bst1, 4 clusters that expressed Ly6c1, two Mcam+ clusters and a cluster of Cd34+ cells. Analysis of the Bst1+ cells revealed FDC-like cells that expressed the marker Cr2 and Cxcl13, and TRC-like cells that expressed Ccl21 and Ccl19. Additional clustering of the Ch25h+ cluster and an Tnfsf13b+ cluster revealed further potential heterogeneity. The authors also examined gene expression in the two subsets of pericyte cells and the small group of CD34+ cells. Finally, they examined the four clusters of closely related Ly6C-expressing cells that expressed genes associated with red pulp fibroblasts.

Closer examination of the Ly6c1+ cells revealed expression of a greater number of ISGs when compared with the other splenic fibroblast subsets. Examination of Stat1-/- mice revealed that many of these ISGs were Stat1-dependent, while increased expression of selected ISGs were also reduced in IFN α 1-/- mice, indicating a role for homeostatic interferon in this gene expression profile. Injection of IFN β into mice stimulated increased Oas1a, Irf7 and Usp18 expression in Ly6c1+ fibroblasts after 2hr. Finally, they showed that splenic fibroblast subsets responded to MCMV infection by increasing ISG expression after 12hr.

The findings presented here are descriptive but are interesting and build on previous studies (references 5 and 6 in particular), especially by adding information about red pulp fibroblast functions. Lacking is biological insight into the role of increased ISG expression by Ly6C+ fibroblasts during infection. In the present form, the study does not show if the signature of ISG expression in red pulp fibroblasts contributes to pathogen control or immunity.

These gaps and omissions will be important to address for publication:

Reviewer #1	Our response
1) The four Ly6c1+ subsets appear to have very similar gene expression. Are these alternate activation states or actual subtypes? Some validation of these clusters is required to ascertain if these are distinct subsets.	All four Ly6c1⁺ FC clusters, i.e., two Hmox1⁻ and two Hmox1⁺ clusters were robustly detected by various clustering algorithms (Leiden, Louvain, SML). In the revised manuscript we present additional flow cytometry-based evidence validating that Ly6C⁺ FC indeed comprise cells with high or low levels of HMOX1 protein (Figure 4d). new Figure 4d:

Manuscript lines 210-211: “Flow cytometric analysis validated the existence of Ly6C⁺ FC expressing high or low levels of HMOX1 protein (Figure 4d), in keeping with the scRNA-seq analysis.”

Due to the high expression of HMOX1 by splenic red pulp macrophages, we were however, not able to reliably identify HMOX1-expressing FC in situ, preventing us from determining if Hmox1⁻ and Hmox1⁺ FC occupy distinct anatomical niches. Regarding the two most similar Ly6c1⁺ FC clusters, termed Ly6c1⁺/Hmox1⁻Ccl5^{lo} and Ly6c1⁺/Hmox1⁻Ccl5^{hi}, we conclude that they likely represent alternative activation states of the same subset, since their relative abundance shifted reciprocally 24 h after viral infection (these cells were not proliferating).

Manuscript lines 302-305: “The relative abundance of individual clusters was similar between both conditions, except for a reciprocal shift between the two most similar Ly6c1⁺ clusters, Ly6c1⁺/Hmox1⁻Ccl5^{lo} and Ly6c1⁺/Hmox1⁻Ccl5^{hi} (these cells were not proliferating), suggesting they may represent alternative activation states of the same subset (Figures 1b and 7d).”

2) In Fig. 1f, Sca-1 is now used to identify cells instead of CD34. Gating should be done with CD34 or the rationale for changing to another marker explained clearly.

In this experiment we specifically addressed the developmental emergence of FC expressing BST-1 or Ly6C markers. Sca-1 was used as an additional maturation marker, replacing CD34 in the panel. We agree with the reviewer that the absence of CD34 in the gating may be confusing. Hence, we repeated this analysis using a panel that includes CD34, replacing the graphs in Figure 1f, which did not influence the results and similarly supported the original conclusions: “Neither BST-1⁺ nor Ly6C⁺ FC were present at birth, but emerged in the first weeks of postnatal life, suggesting that these subsets mature from neonatal precursor population/s (Figure 1f).”

new Figure 1f:

3) The location of the Tnfsf13b+ subclusters needs to be demonstrated to validate if these cells localize to the T-B or B follicles as suggested.

In the revised manuscript we performed histologic examination of the localization of Bst1⁺/Tnfsf13b⁺ FC using in situ RNA hybridization (RNAscope), which is presented in Figure 2f.

new Figure 2f:

This analysis demonstrated that Bst1⁺/Tnfsf13b⁺ FC (identified as cells co-expressing mRNAs for Dpt and Tnfsf13b) are located, not as previously hypothesized at the outskirts of but rather throughout the splenic T cell zone. The existence of Tnfsf13b-expressing FC at the T-B border has previously been demonstrated in the LNs (Cremasco et al. Nature Immunol. 15, 973-981 (2014)), on which premise we built our initial hypothesis about the location of Bst1⁺/Tnfsf13b⁺ FC in the spleen. However, this study did not examine if Tnfsf13b-expressing FC may extend deeper into the T-cell zone, and thus we can no longer compare the distribution of Tnfsf13b-expressing FC between the LNs and the spleen. We have therefore duly removed statements indicating similarity between LN- and splenic Tnfsf13b-expressing FC. We introduced following changes in the manuscript text with respect to the description of Bst1⁺/Tnfsf13b⁺ FC (lines 145-151):

“Histological examination by in situ RNA hybridization (RNAscope) demonstrated that Bst1⁺/Tnfsf13b⁺ FC (identified as cells co-expressing mRNAs for Dpt and Tnfsf13b) reside specifically in the splenic T cell zone (Figure 2f). This data also suggested that Dpt-expressing cells constitute a major source of Tnfsf13b transcripts in the spleen (Figure 2f). Notably, B cell viability and follicular organization in the LNs appear to be maintained by BAFF produced locally by LN FC²⁰. Whether Dpt⁺ FC constitute a biologically relevant source of BAFF in the spleen will require additional functional experiments.”

4) Figure 3b: the CD34 subset is missing from the plot that shows ASMA expression.

CD34⁺ FC were missing from the original analysis because too few events were recorded to reliably address ASMA levels in this lowly abundant subset. We have repeated this experiment with more events recorded and now show results for ASMA expression in all FC subsets in Figure 3b. As expected, CD34⁺ FC expressed less ASMA than MCAM⁺ or BST-1⁺ FC, further corroborating the scRNA-seq analysis shown in Figure 3a. No changes in the manuscript text were necessary.

new Figure 3b:

5) Where are the Cxcl13+Mcam+ localized in the spleen? This should be demonstrated.

In the revised manuscript we validate the existence of MCAM⁺CXCL13⁺ FC using flow cytometry (Figure 3e).

new Figure 3e:

Furthermore, we histologically examine the localization of this subset using in situ RNA hybridization (RNAscope) (Figure 3f).

new Figure 3f:

This analysis indicated that Mcam⁺Cxcl13⁺ FC (identified as cells co-expressing mRNAs for Notch3 and Cxcl13) occupy a restricted perivascular niche.

We introduced following changes in the manuscript text with respect to the description of Mcam⁺Cxcl13⁺ FC (lines 171-180): “Flow cytometric analysis confirmed the distinction of MCAM⁺ FC into MCAM^{hi}CXCL13⁺ and MCAM^{lo}CXCL13⁺ subsets

(Figure 3e), corroborating the scRNA-seq analysis. Next, we assessed the anatomical localization of $Mcam^+|Myh11^{hi}$ FC and $Mcam^+|Cxcl13^+$ FC. To this end, we performed *in situ* RNA hybridization (RNAscope) for *Notch3*, a pericyte/VSMC marker²¹ which robustly discerns both $Mcam^+$ clusters from other FC (Figure 3a), and for *Cxcl13*. In contrast to $Mcam^+|Myh11^{hi}$ FC (identified as $Notch3^+|Cxcl13^-$ cells), which were found in both red and white pulp, $Mcam^+|Cxcl13^+$ FC surrounded select vessels in the red pulp (Figure 3f, filled arrowheads). As far as we are aware, $Mcam^+|Cxcl13^+$ cells have no apparent equivalent among previously described FC populations. Putative specialized function(-s) of the $CXCL13^+$ mural cells in the spleen remain to be addressed by future studies.”

6) The *Wt1* staining in Figure 4 c and d is nice, yet the *Wt1*⁺ cells all appear to be mostly *Meca-32*⁺ endothelial cells. Do the red pulp fibroblasts express *Plvap* that could confound this analysis? Furthermore, endothelial cells in blood vessels and sinusoids express *Ly6C*. It is not clear from these images which cells are fibroblasts and which are endothelial cells, even with the nuclear *Pdgfra-H2B-GFP* reporter. This might be due to the close association of these two cell types, but clearer images are required to demonstrate the cell identities.

We argue that co-expression of *WT1* and *PDGFR α* (as detected by the nuclear *Pdgfra-H2B-GFP* reporter signal) unequivocally identifies *Ly6C*⁺ FC, not EC, as the latter are *PDGFR α* ⁻*WT1*⁻. Supporting evidence is now shown in Supplementary Figures 1d-f.

new Supplementary Figure 1d-f:

We conclude that many but not all *Ly6C*⁺ FC are positioned in direct contact, possibly wrapping around, *MECA-32*⁺ sinusoids. This interpretation is also clearly supported by *Ly6C/desmin/MECA-32* immunostaining shown in revised Figure 4b (note we achieved a better signal for *Ly6C* by optimising tissue fixation method) visualizing the network of *Ly6C*⁺ FC based on co-expression of *Ly6C* and *desmin*. Accordingly, *Ly6C*⁺*desmin*⁺ FC were detected not only in-between *MECA-32*⁺ sinusoids (Figure 4b, filled arrowheads) but also appeared in direct contact with these vessels (Figure 4b, empty arrowheads). In sum, we analysed the localization of *Ly6C*⁺ FC using two methods with concordant results. Given the improved quality of *Ly6C/desmin/MECA-32* analysis in the revised Figure 4b, we removed the analysis of *WT1* alone presented in Figure 4c of the original manuscript since it became redundant.

new Figure 4b:

We introduced following changes in the manuscript text with respect to the histological analysis of *Ly6C*⁺ FC (lines 199-205):

	“Immunohistological staining for a mesenchymal cell marker, desmin and Ly6C demonstrated that Ly6C⁺desmin⁺ FC are selectively located in the red pulp and marginal zone (Figure 4b). Ly6C⁺desmin⁺ FC formed a network in-between MECA-32⁺ sinusoids (Figure 4b, filled arrowheads) but also appeared to tightly wrap around these vessels (Figure 4b, empty arrowheads). The existence of Ly6C⁺ FC positioned in direct contact with MECA-32⁺ endothelium was also apparent when Ly6C⁺ FC were visualized as WT1⁺GFP⁺ cells on splenic sections from Pdgfra-H2B-GFP mice (Figure 4c, filled arrowheads). Of note, EC are PDGFRα WT1⁺ (Supplementary Figure 1d-f).”
7) Of the 104 ISGs that were examined, 46 were higher in Ly6c1+ cells and only 11 in Bst1+ cells. Since these ISGs were identified in lymphoid and myeloid cells, it is possible that this excludes analysis of genes that are regulated by interferon in mesenchymal cell types but not in lymphoid or myeloid cells. Can this be ruled out?	Yes, this was ruled out. Namely, Stat1-dependent enrichment for higher ISG expression in Ly6c1⁺ FC was also observed when the analysis was performed using an ISG set collated for primary fibroblasts, reassuring the conclusions drawn. These results are now included in Supplementary Figure 2b and are discussed in lines 259-263: “Stat1-dependent enrichment for ISG expression in Ly6c1⁺ FC was corroborated using an independent ISG set collated for primary fibroblasts (extracted from the Interferome database v2.0³¹ using the following search criteria: max. 6 hrs post stimulation with IFNβ, (log₂(fold change) > 2; p-val < 0.05) (Supplementary Figure 2b).” new Supplementary Figure 2b: b) ISG set collated for IFN-β-stimulated fibroblasts  Ly6c1⁺ FC vs Bst1⁺ FC wt mice Enrichment score (ES) = 0.34 Normalized ES (NES) = 1.56 Nominal p-value < 0.01 FDR q-value < 0.01 Ly6c1⁺ FC vs Bst1⁺ FC Stat1^{KO} mice Enrichment score (ES) = -0.26 Normalized ES (NES) = -0.837 Nominal p-value = 0.69 FDR q-value = 0.69
8) Amongst the 46 ISG identified in Ly6c1+ cells, not all were Stat1 dependent (figure 6c). They should show how many genes were stat1-dependent, which were or were not, and if the stat1-independent genes were known to be stat1-independent. Similarly, the 11 genes in Bst1+ cells were not found to be stat1-dependent. Was this expected?	In lines 257-259 of the revised manuscript, we now state that “Specifically, Stat1 was responsible for the overexpression of 26 of 46 genes that constituted the ISG signature of Ly6c1⁺ FC but affected none of the 11 ISGs that were overexpressed by Bst1⁺ FC (Supplementary Table 1).” Delineation of which ISGs were differentially expressed in a Stat1-dependent manner and which not, is available in Supplementary Table 1. These results are not unexpected (the steady-state expression of ISGs may or may not be regulated by Stat1 – this depends on the cell type and the strength of tonic IFN signals it receives) and are consistent with Ly6c1⁺ FC being under a greater influence of Stat1-dependent tonic IFN signalling compared to Bst1⁺ FC.
9) Following MCMV infection, all cell types upregulated ISGs, apparently all with a	In the revised manuscript, we duly provide a more precise interpretation and discussion of the analysis of ISG expression upon viral infection stating in lines 301-307 that “Based on the comparison of cZ-scores for ISG expression calculated for individual subsets in the untreated versus infected condition, all FC clusters

similar fold change relative to the homeostatic condition, except for the Cd34+ cells, which appeared to show a greater overall increase in cZ-score. This should be discussed. Despite this graph, the change in expression of ISGs after MCMV infection is not clear at the level of individual genes. The authors conclude: “the ISG signature acquired by Ly6c1+ FC upon MCMV infection was similar to the one expressed by these cells in homeostatic conditions”. A depiction of this data is required such as a heatmap of the uninfected versus infection condition to demonstrate how individual ISG expression is altered by infection.

upregulated ISGs following MCMV infection, with the highest overall ISG expression noted for Ly6c1+ FC and for a fraction of Cd34+ FC (Figure 7f). A more detailed evaluation on the level of individual ISGs revealed that even though Cd34+ FC reached a similar ISG score, Ly6c1+ FC were the only FC subset uniquely overexpressing a sizeable array of ISGs in virus-infected mice (Figure 7g).” Furthermore, in revised Figure 7g we present data which more clearly show that ISGs selectively overexpressed by Ly6c1+ FC in virus-infected mice overlap with ISGs overexpressed by these cells in the steady state. This figure now also depicts the average expression level of individual ISGs before and after infection (indicated to the right of each panel).

10) To what degree does the time point of the analysis after MCMV infection (12hr) influence these results? This might be expected to differ in the different cell types as the infection progresses and potentially change this pattern of expression.

In the revised manuscript we “complemented the scRNA-seq analysis performed at 24 h post infection with a time-resolved profile of ISG expression by splenic FC in virus-infected mice. To this end, we performed a kinetic analysis of the expression of Irf7 and Oas1a, which are overexpressed in Ly6c1+ FC at 24 h post MCMV infection (Figure 7g), in Ly6c1+ FC and Bst1+ FC at 12, 36 and 48 h post infection with 10⁶ PFU of MCMV i.p. by RT-qPCR. The ISG response of splenic FC was biphasic, matching the biphasic kinetic of type I IFN production upon MCMV infection, which peaks in the first 24 h and then again at 48 h post infection³³. Importantly, Ly6c1+ FC expressed the tested ISGs at a higher level compared to Bst1+ FC, with highest differences at 12 h and at 48 h (Figure 7i).” (lines 314-322).

	i) Normalized expression hours post infection Ly6c1⁺ FC (red triangles) Bst1⁺ FC (blue squares) Irf7 Oas1a
11) Is the ISG response to MCMV stat1-dependent?	This question was not in the focus of our study. We analysed the immune response of splenic FC to virus infection. We do not try to make claims about which signalling pathways and/or soluble cytokine mediators regulate ISG expression in FC upon virus infection. We can speculate (speculation not included in the manuscript) that ISGs in virus infected mice are regulated by the superposition of several distinct cytokine pathways, amongst which type I IFNs may be dominant (since the ISG response in splenic FC kinetically follows the waves of type I IFN production in MCMV-infected mice (Figure 7i)). Given this and the importance of Stat1 in mediating the canonical ISG response to acutely produced IFNs it is likely that the ISG response of splenic FC to MCMV is, at least partially, Stat1-dependent.
12) What is the biological relevance of increased ISG expression by Ly6C+ cells during homeostasis? Does the proposed innate immune function of splenic red pulp stroma provide some degree of protection from infection that the authors can demonstrate? Such biological insight is currently missing from these observations.	Indeed, we agree with the reviewer that “the putative importance of Ly6c1⁺ FC in innate antiviral defence remains to be directly addressed by future studies using conditional knockout models” (lines 325-326). We have, in fact, tried to answer these important outstanding questions in the course of this study, but ultimately could not do it because the Ly6C⁺ FC-specific Cre-driver line, Tcf21-Cre^{ER} (Inra et al. Nature 527, 466-471, (2015)), which we chose did not provide adequate deletion efficiency for the floxed alleles of Stat1 or Ifnar1. On top of it the Tcf21-Cre^{ER} strain turned out to be generally unsuitable for studies of innate immunity in the spleen since we observed that mice carrying solely the Tcf21-Cre^{ER} knock-in allele are considerably more susceptible to virus infection of the spleen, likely resulting from the severe reduction in the number of splenic macrophages in these mice.

Reviewer #2 Immunology and Cell Biology, immune cells development

This manuscript carefully details the molecular analyses of stromal cells in the spleen. While this is not the first study to endeavor to do this, the heterogeneity and complexity of this tissue framework results in new discoveries building on previous evidence. In this study, the authors identify a number of distinguishing features of different subsets of stromal cells and go on to undertake some cellular characterization of these cells. The authors go on to focus on the Ly6c1+ subset and investigate the origins of the ISG regulated pathway to discover that Stat1 was key to induction and maintenance of this pathway in Ly6c1+ FC.

Major points:

1) The authors note that Ly6C is reduced in expression in Stat1 deficient cells. How can they confirm that they	We thank the reviewer for alerting us that the gating strategy was not clearly explained. In the revised manuscript we clarify this in lines 251-253: “To ensure correct subset identification (note Ly6c1 is a type I IFN-inducible gene³⁰ expressed by Ly6c1⁺ FC of Stat1^{KO} mice at a modestly reduced level), Ly6c1⁺ FC were sorted as MCAM-CD34-BST-1-PDGFRα⁺ FC (Supplementary Figure 2a).”
---	---

then recover the same population as they might identify in wildtype mice and that the low FCs are also not part of the group that are involved? Is there an independent marker that could be used to confirm this?

new Supplementary Figure 2a

2) Given that following MCMV infection all FC upregulate the antiviral gene signature, what is the contribution of these lower expressing subsets to immune protection.

This question was not in the focus of our study. At this stage we can only speculate (speculation not included in the manuscript) that Ly6C⁺ FC are “special” in that they may be able to withstand a higher virus burden than other FC subsets, the latter mounting a “normal” ISG response protecting against a lower viral dose.

3) Given that Ly6C⁺ cells are preferentially infected but excluded from the dataset, how does this skew the data and the interpretation.

We thank the reviewer for raising the issue that the removal of infected cells was not clearly explained. This information was duly included in the revised manuscript (lines 293-297): “In keeping with the flow cytometric analysis (Figure 7c), MCMV-infected cells (3.1 %) resided in a distinct cluster that expressed markers of cellular stress and clustered closely with Ly6c1⁺ FC (Supplementary Figure 3a). Virus-infected cells were removed from subsequent analysis as we aimed to assess the bystander immune response of FC and not their cell-intrinsic response to the virus.” Given that virus-infected cells constituted a very small fraction of FC, their removal could not skew the data or the interpretation.

new Supplementary Figure 3a

4) Was protein expression examined in the Bst1+ FC?

The expression of BST-1 in the Bst1⁺ FC was validated in the original manuscript (Figure 1e). In addition to this: “In keeping with published data on the expression pattern of BST-1 (CD157/BP-3) in the spleen¹⁵, we found that Bst1⁺ FC (identified as BST-1⁺ cells co-expressing a mesenchymal cell marker, desmin) are selectively localized in the white pulp (Figure 2a).” (lines 119-121)

new Figure 2a

Figures:

1. The number of replicates for cellular experiments has not been indicated, nor whether the data are representative/pooled, and how many animals were analysed.	We thank the reviewer for alerting us that this information was missing. It has now been duly detailed in Figure Legends. Below we present updated descriptions for all cellular experiments: Figure 1e “Flow cytometric confirmation of FC subsets in the adult spleen. Representative stains from 2 independent experiments with 3 biological replicates per experiment using cell preparations from a single spleen/replicate.” Figure 1f “Analysis of the emergence of FC subsets during spleen ontogeny. Representative stains from 2 independent experiments with 3 biological replicates per experiment using pooled cell preparations from 2 spleens/replicate.” Figure 2c “Flow cytometric confirmation of BST-1⁺ TRC and FDC. Representative stains from 2 independent experiments with a single biological replicate per experiment using pooled cell preparations from 2 spleens/replicate.” Figures 3b-c “Flow cytometric analysis of b) ASMA or c) PDGFRα expression by the indicated FC subsets. Representative stains from 2 independent experiments with a single biological replicate per experiment using pooled cell preparations from 2 spleens/replicate.” Figure 3e “Flow cytometric analysis of CXCL13 expression by the indicated FC subsets. Representative stains from 2 independent experiments with 2 biological replicates per experiment using cell preparations from a single spleen/replicate.” Figure 4d “Flow cytometric analysis of HMOX1 expression by the indicated FC subsets. Representative stains from 2 independent experiments with 2 biological replicates per experiment using cell preparations from a single spleen/replicate.” Figure 6f “RT-qPCR analysis of ISG expression in Ly6c1⁺ FC and Bst1⁺ FC isolated from the spleens of wt and Ifnar1^{KO} mice. Data are pooled from 2 independent experiments and presented as arithmetic mean \pm SD of 3 biological replicates (depicted as symbols) using pooled cells from 2 mice/replicate.” Figure 7a “Data from singular in vivo experiment is presented as arithmetic mean \pm SD of 3 biological replicates (depicted as symbols) using pooled cells from 2 mice/replicate.” Figure 7b “Data from singular dose-response experiment with one mouse per each indicated IFN dose. Symbols depict ISG expression in indicated subsets sorted from a single mouse/IFN dose. Figure 7c “Percentage of GFP⁺ cells among indicated splenic subsets 12 h post infection of mice with 10⁶ PFU of MCMV^{GFP} i.p. Data from one representative
--	---

	experiment of two independent experiments is presented as arithmetic mean \pmSD of 4 biological replicates (depicted as symbols) using cell preparations from a single spleen/replicate.” Figure 7i “Kinetic assessment of ISG expression in Ly6c1⁺ FC and Bst1⁺ FC following infection of mice with 10⁶ PFU of MCMV i.p. determined by RT-qPCR. Data from singular in vivo experiment is presented as arithmetic mean \pmSD of 3-4 biological replicates (depicted as symbols) using cells sorted from a single spleen/replicate.”
2. The authors have several instances where they skip over figures in their reporting eg. p 6: (Fig. 2a and Fig. 4a), similarly in skipping figures to refer to Fig. 7.	We concur with the reviewer that this was confusing and accordingly, refrained from skipping figures in the revised manuscript.

Reviewer #3 Bioinformatics, Single-Cell Genomics

In their manuscript, Pezoldt et al. investigate the heterogeneity of splenic fibroblasts in the mouse using a variety of methods. They identify four main populations of cells, as well as a number of subclusters, and characterize their response to viral infection and stimulation with type I interferons. This is a well-performed study with experiments nicely complementing each other. However, there are some weak points especially with respect to the interpretation of results that should be addressed prior to acceptance of the manuscript for publication.

1) Little evidence is presented that the clusters identified really represent meaningfully different cell types or states. How was the resolution of clustering determined? I assume that FindMarkers was used with default setting, i.e. Louvain clustering. Did the authors try a different approach such as Leiden clustering? Did the authors perform stainings to assess whether the different subpopulations/states really exist?	In the revised manuscript we provide the following validation data for the newly identified FC subsets: I) using in situ RNA hybridization, we demonstrate that Bst1⁺/Tnfsf13b⁺ FC localize in the splenic T cell zone (Figure 2f). new Figure 2f  f) Tnfsf13b Dpt Cxcl13 DAPI We introduced following changes in the manuscript text with respect to the description of Bst1⁺/Tnfsf13b⁺ FC (lines 145-151):
---	---

“Histological examination by *in situ* RNA hybridization (RNAscope) demonstrated that $Bst1^+/Tnfsf13b^+$ FC (identified as cells co-expressing mRNAs for *Dpt* and *Tnfsf13b*) reside specifically in the splenic T cell zone (Figure 2f). This data also suggested that *Dpt*-expressing cells constitute a major source of *Tnfsf13b* transcripts in the spleen (Figure 2f). Notably, B cell viability and follicular organization in the LNs appear to be maintained by BAFF produced locally by LN FC²⁰. Whether Dpt^+ FC constitute a biologically relevant source of BAFF in the spleen will require additional functional experiments.”

2) using flow cytometry, we confirm the existence of $MCAM^+CXCL13^+$ FC (Figure 3e)

new Figure 3e

and demonstrate that this subset is positioned around select vessels in the red pulp (Figure 3f)

new Figure 3f

We introduced following changes in the manuscript text with respect to the description of $Mcam^+|Cxcl13^+$ FC (lines 171-180): “Flow cytometric analysis confirmed the

distinction of MCAM⁺ FC into MCAM^{hi}CXCL13⁻ and MCAM^{lo}CXCL13⁺ subsets (Figure 3e), corroborating the scRNA-seq analysis. Next, we assessed the anatomical localization of Mcam⁺/Myh11^{hi} FC and Mcam⁺/Cxcl13⁺ FC. To this end, we performed in situ RNA hybridization (RNAscope) for Notch3, a pericyte/VSMC marker²¹ which robustly discerns both Mcam⁺ clusters from other FC (Figure 3a), and for Cxcl13. In contrast to Mcam⁺/Myh11^{hi} FC (identified as Notch3⁺Cxcl13⁻ cells), which were found in both red and white pulp, Mcam⁺/Cxcl13⁺ FC surrounded select vessels in the red pulp (Figure 3f, filled arrowheads). As far as we are aware, Mcam⁺Cxcl13⁺ cells have no apparent equivalent among previously described FC populations. Putative specialized function(-s) of the CXCL13⁺ mural cells in the spleen remain to be addressed by future studies.”

4) using flow cytometry, we validate that Ly6C⁺ FC are comprised of cells with high or low levels of HMOX1 protein (Figure 4d).

new Figure 4d

Manuscript lines 210-211: “Flow cytometric analysis validated the existence of Ly6C⁺ FC expressing high or low levels of HMOX1 protein (Figure 4d), in keeping with the scRNA-seq analysis.”

While in the paper we show clustering done using the Louvain algorithm, the clusters described were also detected by other clustering methods, such as Leiden (shown below, not included in the manuscript) and SML (yielded identical results). Note that Bst1⁺/Tnfsf13b⁺ FC are recovered from the initial Leiden clustering following re-embedding of Bst1⁺/Ccl21a⁺ FC and Cd34⁺ FC.

Leiden clustering

Re-embedding of Cd34⁺ and Bst1⁺|Ccl21a⁺ clusters

2) Since the main populations aren't very well defined on the UMAP, it would be great to see a UMAP colored by count matrix to be able to assess the quality of integration.

Indeed, this is a valuable metric to evaluate the quality of the scRNA-seq data and the quality of the integration. We therefore have now included the number of UMIs per cell, the percentage of reads mapping to heat-shock protein genes, the percentage of mitochondrial reads and the percentage of reads mapping to ribosomal protein genes (Supplementary Figure 1b). As expected, we do observe that the main cell types differ with regard to, for example, mitochondrial transcripts and number of UMIs per cell, which does not extend to the sub-clusters for the given cell type. Therefore, we conclude that these features are not a confounder for the identification of FC populations within the main cell types.

Supplementary Figure 1b

3) Concerning this, did the authors get similar results when analyzing the non-infected data alone, without integrating data from both experiments? What was the distribution of cell types between the different groups?

Yes, consistent results were obtained for most of the previously delineated clusters, when analyzing the non-infected data alone (shown below, not included in the manuscript).

4) Cells containing viral reads were excluded from the analysis. I assume this was done to investigate the reaction of bystander cells rather than the cell-intrinsic immune response. In that case, however, it would be relevant to see the distribution of viral reads across all cells. If this is not clearly bimodal, i.e. viral UMI counts are low, no clear separation is possible between infected and non-infected cells. In line with this, which subpopulations of FCs tended to be infected?

Removal of virus-infected cells was indeed done to assess the bystander immune response of FC and not the cell-intrinsic response to the virus. Based on analysis of the distribution of viral reads (shown below, not included in the manuscript) we identified a discrete subset of cells (3.1%) highly enriched for viral reads (see red cut-off of 1% viral UMI) that we considered virus-infected and that were removed from subsequent analysis. As is now shown in Supplementary Figure 3a, these cells resided in a distinct cluster that expressed markers of cellular stress and clustered closely with Ly6c1⁺ FC, fitting the characteristics of virus-infected FC determined by flow cytometry in Figure 7c, which were mainly Ly6C⁺.

As can be further seen from the distribution of viral reads there was also a fraction of cells (ca 5%) with distinctively lower counts of viral reads, possibly a result of contamination with ambient viral RNA, which were not eliminated from the analysis. Importantly, as shown above (graph to the right), cells with low abundance of viral reads distributed evenly across the main FC clusters, ruling out the possibility that they could significantly influence the results or conclusions.

new Supplementary Figure 3a:

Virus-infected cells are now discussed in the revised manuscript in lines 293-297 as follows: “In keeping with the flow cytometric analysis (Figure 7c), MCMV-infected cells (3.1 %) resided in a distinct cluster that expressed markers of cellular stress and clustered closely with Ly6c1⁺ FC (Supplementary Figure 3a). Virus-infected cells were removed from subsequent analysis as we aimed to assess the bystander immune response of FC and not their cell-intrinsic response to the virus.”

5) Figure 7f does not really support the statement that ISG-upregulation is specific to the Ly6c1⁺ subpopulation, as Cd34⁺ FCs appear to show an upregulation to similar levels at least in part of the cells.

This has been now corrected and accordingly rewritten as follows (lines 305-314): “Based on the comparison of cZ-scores for ISG expression calculated for individual subsets in the untreated versus infected condition, all FC clusters upregulated ISGs following MCMV infection, with the highest cZ-scores noted for Ly6c1⁺ FC and for a fraction of Cd34⁺ FC (Figure 7f). A more detailed evaluation on the level of individual ISGs revealed that even though some Cd34⁺ FC reached a similar ISG cZ-score as Ly6c1⁺ FC, Ly6c1⁺ FC were the only FC subset uniquely overexpressing a sizeable array of ISGs in virus-infected mice (Figure 7g). Further importantly, Ly6c1⁺ FC in virus-infected mice were, like in the steady state, selectively enriched for antiviral gene expression, as underscored by gene ontology analysis (Figure 7h).”

6) For the qPCR experiments: the normalized expression seems to indicate that the expression of some targets is orders of magnitude lower than that of the housekeeping gene used and varies by similar levels between different conditions. Were primer efficiencies in the corresponding expression ranges determined to confirm that ddCt can be used rather than measuring a standard curve?

Relative gene expression was calculated with normalization to the expression of Gapdh. Primers were validated for the use of the ddCt method by determining the efficiency of each primer pair in the respective expression range. Primer validation data (not included in the manuscript) are shown below:

7) Could the authors give some details about how GSEA was performed, specifically how the genes were filtered, what ranking metric was used, and in which mode GSEA was run (preranked or standard)?	We thank the reviewer for alerting us about this omission. Description of gene set enrichment analysis is now duly provided in the Methods section (lines 482-487): “Gene set enrichment analysis was performed in the pre-ranked mode using the GSEA desktop application v4.1.0 (https://www.gsea-msigdb.org/gsea). Analysis was performed on DEGs ($\log_2(\text{fold change}) > 0.8$, $p\text{-val} < 0.05$) from the comparison between Ly6c1^+ FC and Bst1^+ FC from the wild-type or Stat1^{KO} condition with fold change serving as the ranking metric. The number of permutations was set to 1,000“
--	--

Reviewers' comments:

Reviewer #1 (Remarks to the Author):

The authors have significantly improved the manuscript by adding new data and analysis. The RNAscope validation images are excellent. The presence of CXCL13-expressing MCAM+ cells in the red pulp is interesting. This study is important and should be published, but there are just a few small remaining concerns:

- The RNAscope data in Fig. 2F does not appear to show CXCL13 expression in RP, unlike Figure 3F. In how many mice was CXCL13 observed in the red pulp, especially around MCAM+ cells?
- The flow cytometry analysis of CXCL13+ MCAM+ cells was not very convincing. Some quantitation of this data across multiple samples would be helpful to demonstrate if this population is consistent.
- Likewise, for the Hmox1 staining by flow cytometry, the authors should show data from multiple mice. This is important because by scRNAseq the Hmox1+ cells were about half of the Ly6c1+ cells, but only a small part of the population by flow cytometry.
- Ly6C+Hmox1+C1q+ cells expressed many genes also found in macrophages (ie Fcgr1g, C1q, Hmox1, Cd68, Spi1, Spic). Can contamination by macrophages be excluded?

Reviewer #2 (Remarks to the Author):

The authors have comprehensively responded to the issues raised by the reviewers. This significantly strengthens the work.

Reviewer #3 (Remarks to the Author):

The authors have addressed my concerns from the previous review round, and for me, no further concerns arise from their response. I thus recommend acceptance of the manuscript for publication.

"Single-cell transcriptional profiling of splenic fibroblasts reveals subset-specific innate immune signatures in homeostasis and during viral infection"

Dear reviewers,

We thank the reviewers for the valuable feedback. We have addressed the four additional minor remarks by Reviewer 1 in a point-by-point reply and provide a revised manuscript that is amended with the requested analyses that further strengthen its conclusions. In the article file, changes are highlighted in yellow.

Reviewers' comments:

Reviewer #1 (Remarks to the Author):

The authors have significantly improved the manuscript by adding new data and analysis. The RNAscope validation images are excellent. The presence of CXCL13-expressing MCAM+ cells in the red pulp is interesting.

This study is important and should be published, but there are just a few small remaining concerns:

Reviewer #1	Our response
1) The RNAscope data in Fig. 2F does not appear to show CXCL13 expression in RP, unlike Figure 3F. In how many mice was CXCL13 observed in the red pulp, especially around MCAM+ cells?	Cxcl13⁺Notch3⁺ FC were observed in the red pulp consistently in all sections analysed with the Cxcl13/Notch3 RNAscope panel (i.e., 2 spleens analysed in technical duplicates across 2 independent experiments). As we clarify now in the manuscript (lines 177-181): "Of note, the Cxcl13 RNAscope performed in conjunction with the detection of Dpt and Tnfrsf13b shown in Figure 2f serves only to reveal the positioning of B cell follicles and should not be used to assess the distribution of Cxcl13⁺ FC due to the markedly lower sensitivity of Cxcl13 detection relating to the use of a weaker fluorophore and a detection channel with a higher autofluorescence level."
2) The flow cytometry analysis of CXCL13+ MCAM+ cells was not very convincing. Some quantitation of this data across multiple samples would be helpful to demonstrate if this population is consistent.	The MCAM⁺CXCL13⁺ population was consistent across all samples analyzed (i.e., 4 mice analyzed in 2 independent experiments). Data from individual biological replicates are shown below:  A clearer gating for the CXCL13-expressing Mcam⁺ FC and quantification of the percentage of CXCL13⁺ cells amongst Mcam⁺ FC have been duly included in the manuscript in Figures 3e-f: new Figures 3e-f:

Figure legend: “**e-f**) Flow cytometric analysis of CXCL13 expression. Numbers are percentage of cells in the indicated gates. **e**) Representative stains and **f**) percentage of CXCL13⁺ cells amongst Mcam⁺ FC. Data are pooled from 2 independent experiments and presented as arithmetic mean +/- SD of 4 biological replicates (depicted as symbols) using cell preparations from a single spleen/replicate.”

3) Likewise, for the Hmox1 staining by flow cytometry, the authors should show data from multiple mice. This is important because by scRNAseq the Hmox1⁺ cells were about half of the Ly6c⁺ cells, but only a small part of the population by flow cytometry.

The HMOX1 staining pattern was consistent across all samples analyzed (i.e., 4 mice analyzed in 2 independent experiments). Data from individual biological replicates are shown below:

Gating and quantification of the percentage of HMOX1^{hi} cells amongst Ly6c⁺ FC have been duly included in the manuscript in Figures 4d-e: new Figures 4d-e:

Figure legend: “**d-e**) Flow cytometric analysis of HMOX1 expression. Numbers are percentage of cells in the indicated gates. **e**) Representative stains and **f**) percentage of HMOX1^{hi} cells amongst Ly6c1⁺ FC. Data are pooled from 2 independent experiments and presented as arithmetic mean +/- SD of 4 biological replicates (depicted as symbols) using cell preparations from a single spleen/replicate.”

We included a more detailed description of the above flow cytometry data in the manuscript text (lines 220-223):

“Flow cytometric analysis validated the existence of Ly6c1⁺ FC expressing high or low levels of HMOX1 protein (Figure 4d). The fractional abundance of HMOX1^{hi} FC was consistent with the frequency of the Ly6c1⁺/Hmox1⁺C1q⁺ subcluster expressing the highest level of Hmox1 mRNA (Figures 4a and 4e).”

4) Ly6C+Hmox1+C1q+ cells expressed many genes also found in macrophages (ie Fcgr1g, C1q, Hmox1, Cd68, Spi1, Spic). Can contamination by macrophages be excluded?

Yes, contamination by macrophages was excluded as Ly6c1⁺/Hmox1⁺C1q⁺ FC were devoid of other mRNAs abundantly expressed by splenic red pulp macrophages (for example, *Adgre1* (encoding F4/80), *Mertk*, *Fcgr1* (encoding CD64) and *Csf1r*) or by marginal zone-/marginal metallophilic macrophages (for example, *Itgam* (encoding CD11b), *Siglec1* (encoding CD169), *Marco*, and *Cd209b* (encoding SIGNR1)). Further consistently, Ly6c1⁺/Hmox1⁺C1q⁺ FC clustered with other Ly6c1⁺ FC and expressed stromal cell-specific markers characterizing Ly6c1⁺ FC (for example, *Wt1* and *Tcf21*) at similar levels compared to other Ly6c1⁺ FC subclusters. Thus, Ly6c1⁺/Hmox1⁺C1q⁺ FC have fibroblastic cell identity. It remains possible that Ly6c1⁺/Hmox1⁺C1q⁺ FC acquire select macrophage-derived mRNAs in a process of extracellular vesicle exchange. Whether this is true or not remains outside the scope of the current study.

Reviewer #2 (Remarks to the Author):

The authors have comprehensively responded to the issues raised by the reviewers. This significantly strengthens the work.

Reviewer #3 (Remarks to the Author):

The authors have addressed my concerns from the previous review round, and for me, no further concerns arise from their response. I thus recommend acceptance of the manuscript for publication.

REVIEWERS' COMMENTS:

Reviewer #1 (Remarks to the Author):

The authors have addressed the remaining concerns and improved the manuscript. I recommend publication of this excellent work.